# BaTwa populations from Zambia retain ancestry of past hunter-gatherer groups

Gwenna Breton [1,2] ✉, Lawrence Barham[3], George Mudenda[4,5], Himla Soodyall[6,7], Carina M. Schlebusch [1,8,9,10] & Mattias Jakobsson [1,8,9,10] ✉

Sub-equatorial Africa is today inhabited predominantly by Bantu-speaking groups of Western African descent who brought agriculture to the Luangwa valley in eastern Zambia ~2000 years ago. Before their arrival the area was inhabited by hunter-gatherers, who in many cases were subsequently replaced, displaced or assimilated. In Zambia, we know little about the genetic affinities of these hunter-gatherers. We examine ancestry of two isolated communities in Zambia, known as BaTwa and possible descendants of recent hunter-gatherers. We genotype over two million genome-wide SNPs from two BaTwa populations (total of 80 individuals) and from three comparative farming populations to: (i) determine if the BaTwa carry genetic links to past hunter-gatherer-groups, and (ii) characterise the genetic affinities of past Zambian hunter-gatherer-groups. The BaTwa populations do harbour a hunter-gatherer-like genetic ancestry and Western African ancestry. The hunter-gatherer component is a unique local signature, intermediate between current-day Khoe-San ancestry from southern Africa and central African rain-forest hunter-gatherer ancestry.

Africa is the cradle of modern humans, and the continent is characterised by high genetic, linguistic, and cultural diversity[1,2]. Nevertheless, the majority of populations in sub-equatorial Africa are Bantu-speaking farmers of western African descent, who display genetic, linguistic and cultural homogeneity[3-6] compared to other groups. This homogeneity results from a large-scale movement of people which started ~5000–4000 years ago in western Africa and resulted in a diffusion of languages and people to many parts of the African continent[4,7-9]. This migration was at least partially associated with the spread of agricultural practices (especially at the later stages), and reached eastern Africa and later southern Africa in the last 3000–1000 years[2,6,10].

In many parts of Africa, including central and southern Africa, the incoming Bantu-speakers replaced, dispersed, or assimilated indigenous forager groups. Some groups who traditionally are, or were, hunter-gatherers (HG) have retained specific traits that distinguish them from their neighbours, such as language, subsistence practices, knowledge of their environment and distinct cultural practices. The best-known African hunter-gatherer populations include the rainforest hunter-gatherers from central Africa (thereafter RHG), the Khoe-San from southern Africa (KS), and the eastern African hunter-gatherers (eHG), such as the Hadza. These populations are genetically distinct from neighbouring Bantu-speaking populations of western African descent[7,11-13]. Today these groups have small census sizes and are

[1]Department of Organismal Biology, Human Evolution, Evolutionary Biology Centre, Uppsala University, Uppsala, Sweden. [2]Department of Clinical Genetics and Genomics, Centre for Medical Genomics, Sahlgrenska University Hospital, Gothenburg, Sweden. [3]Department of Archaeology, Classics & Egyptology, University of Liverpool, Liverpool, UK. [4]Livingstone Museum, Livingstone, Zambia. [5]National Museums Board, Lusaka, Zambia. [6]Division of Human Genetics, School of Pathology, Faculty of Health Sciences, University of the Witwatersrand and National Health Laboratory Service, Johannesburg, South Africa. [7]Academy of Science of South Africa, Pretoria, South Africa. [8]Palaeo-Research Institute, University of Johannesburg, Auckland Park, South Africa. [9]SciLifeLab, Uppsala, Sweden. [10]These authors jointly supervised this work: Carina M. Schlebusch, Mattias Jakobsson ✉e-mail: gwenna.breton@gu.se; mattias.jakobsson@ebc.uu.se

dispersed over sub-Saharan Africa; however, it is likely that in the past, there were many more such populations[14]. A way to learn about these past local populations is to obtain ancient DNA from relevant material[15–21]. Another way is through studying the genomes of minority (present-day) groups from under-studied areas that may have ancestry linked to the past groups[22]. Linguistic studies and ethnology help to hypothesise which populations might have retained local ancestry and identify candidate populations for including in genome-wide investigations of the pre-Bantu-speakers history of sub-equatorial Africa.

Despite increasing sequencing efforts of African populations, some areas or populations remain under-represented; one such area is southcentral Africa, which is currently predominantly peopled by Bantu-speaking agropastoralist populations. Morphological studies of skeletal remains from southcentral Africa (Malawi and Zambia) suggest that Middle and Late Holocene (5000 to 500 years ago) groups were more similar to western Africans or to RHG than to KS populations, and that there was no major change between Middle and Late Holocene populations[23–25].

The Luangwa valley in eastern Zambia is particularly informative about past interactions between farmers and foragers in the region[26]. There is evidence for the arrival of food production in the valley from 400 AD, corresponding to the southward spread of the Chifumbaze archaeological complex[27]. Archaeological research points to an even earlier arrival of farmers ~100 AD and an extended overlap of local HG and incoming farmer communities until ~1700 AD[28], after which there is no evidence of a HG presence[28]. On the high escarpment bordering the valley, HGs co-existed with farming communities until the early 19th century AD[29,30], and a similar pattern of extended coexistence is evident in adjacent areas of northern Zambia[29,31,32]. Historical accounts also indicate a late 19th century HG presence on the margins of the Luangwa valley[33]. Such a long coexistence might have been facilitated by the fertile soils and abundance of game in the valley that enabled the two communities to maintain relative independence[29]. The prevalence of tsetse fly in the area may have also limited the spread of agropastoralists contributing to the persistence of HG populations[26].

There are no remaining HG groups in the Luangwa valley region, but the local farmer communities, the Bisa and the Kunda, mention the presence of HGs in their oral history[34]. The mitochondrial DNA and the Y chromosome of the Bisa and Kunda were investigated and most of the diversity was typical of Bantu-speaking populations[34]. The uniparental markers of other Zambian populations were investigated with similar results[35,36]. However, these observations do not rule out the possibility that other southcentral African populations have ancestry from past forager groups which were not similar to extant central and southern African HGs, or that they did not differ enough from present-day Bantu-speakers to be able to distinguish them based on uniparental loci alone. In fact, certain groups in Zambia have been distinguished from Bantu-speaking farmers based on uniparental markers and click sounds in their language; in particular, the Fwe from southwestern Zambia carry ~25% of Khoe-San maternal lineages[37]. This suggests that a few specific minority populations might retain more local ancestry, though the majority does not.

In neighbouring Mozambique, a study of autosomal diversity in 12 Bantu-speaking populations showed that they had little to no admixture with hypothetical aboriginal HG groups[38]. An ancient DNA study presented similar conclusions for Malawi, showing that there has been a recent population replacement, with individuals who lived between 2000 and 8000 years ago genetically resemble HGs, and more recent individuals resemble Bantu-speaking farmers. These mid-late Holocene Malawi HGs were inferred to have ancestry in-between eHGs and current day southern African KS, plus a modest fraction of RHG ancestry[17].

Communities known locally as BaTwa (Ba is the plural prefix when referring to people -Twa- in Bantu languages[39]) are of particular interest when looking for a hypothetical local HG ancestry in Zambia. The term BaTwa is a generic label meaning "people who always move" or "the others", and is applied to groups of foragers and fishers found from the Great Lakes region southward to central Zambia[40]. Early genetic research[41,42] identified a distinctive group, the rainforest hunter-gatherers (referred to as Pygmies in the work of Cavalli-Sforza and colleagues), with this attribution extended to little known groups from southcentral Africa beyond the rainforest margins, such as the Vatwa and the Bambote[43] as well as the BaTwa of Zambia (though no genetic data was available for the BaTwa)[42].

In central Zambia, BaTwa communities are found on the Kafue floodplain (flats)[44,45] and in the Lukanga swamp[46,47]. In northern Zambia they live in and around the Lake Bangweulu wetlands[48,49]. The BaTwa from the Kafue Flats are isolated fishing and hunting communities living on islands, and more recently some also practice agriculture[50]. They are Chitwa-speakers (a dialect of Tonga or Ila Bantu languages)[50]. The BaTwa from the Bangweulu wetlands are fishing and hunting communities, and speak the same Bantu language, ChiBemba, as their neighbours[49]. The neighbours of both communities identify them as different, even though they speak the same language, and they live as marginalised groups[49,51]. The communities themselves are reluctant to discuss their past with outsiders and they are hesitant to self-identify as BaTwa in the presence of Bantu-speaking farmers[40,49]. In a linguistic survey of the BaTwa from the Kafue Flats[50], it is reported that the BaTwa and the Ila (a neighbouring group) see themselves as indigenous to Zambia, contrary to the other neighbouring groups, such as the Lozi, who trace their origins to other countries such as Angola or the Democratic Republic of the Congo (DRC). The BaUnga from Bangweulu wetlands, who are related to the Bemba, report that they found BaTwa when they arrived from the DRC[48,49]. Little more is known about the distant past of the BaTwa.

To address the issue of the possible local hunter-gatherer ancestry of the BaTwa[45,48] we generated genome-wide SNP data for 40 individuals each from two BaTwa populations, the BaTwa from Lake Bangweulu and the BaTwa from Kafue Flats near the Lochinvar National Park (Fig. 1a). We made use of the Illumina H3Africa array, designed specifically to capture African genetic diversity[52]. To put the BaTwa genetic diversity in context of neighbouring populations, we analysed the data together with individuals from three Zambian Bantu-speaking agropastoralist populations, the Bemba, Lozi, and Tonga[53]. We describe the autosomal genetic diversity of these Zambian populations in the context of worldwide genetic diversity, with a focus on sub-Saharan African populations. We also characterise their demographic history, including past admixture events.

## Results

### Genetic affinities of the Zambian populations

We explored population relations among the five Zambian populations (Fig. 1a, Table 1) together with a diverse set of African populations and two populations from outside Africa, in a dataset containing 344,644 variants and 973 individuals (from 39 populations, Fig. 1b, Supplementary Table 1). The first axis in a principal component analysis (PCA) separates the African populations from the non-African populations (Supplementary Fig. 1). The subsequent axes separate different African groups. We projected the five Zambian populations on a PCA of three populations that represent the genetic diversity of sub-Saharan Africa: the Ju|'hoansi (KS), the Yoruba (Niger-Congo speakers from western Africa) and the Baka from Cameroon (RHG) (Fig. 1c). In this PCA space, the BaTwa individuals form a cline, from the west African Yoruba (roughly starting from the Bemba, Lozi, and Tonga individuals), towards a space in-between the Ju|'hoansi and Baka. The individuals from Kafue are further away from the Bemba, Lozi, and Tonga than those from Bangweulu. The cline runs somewhat deflected towards the Baka cluster of the triangle, suggesting some level of intermediate relationship with the Baka and the Ju|'hoansi. The three Zambian agropastoralist populations group together, close to the BaTwa, but are offset more towards the Yoruba. This suggests that the Zambian

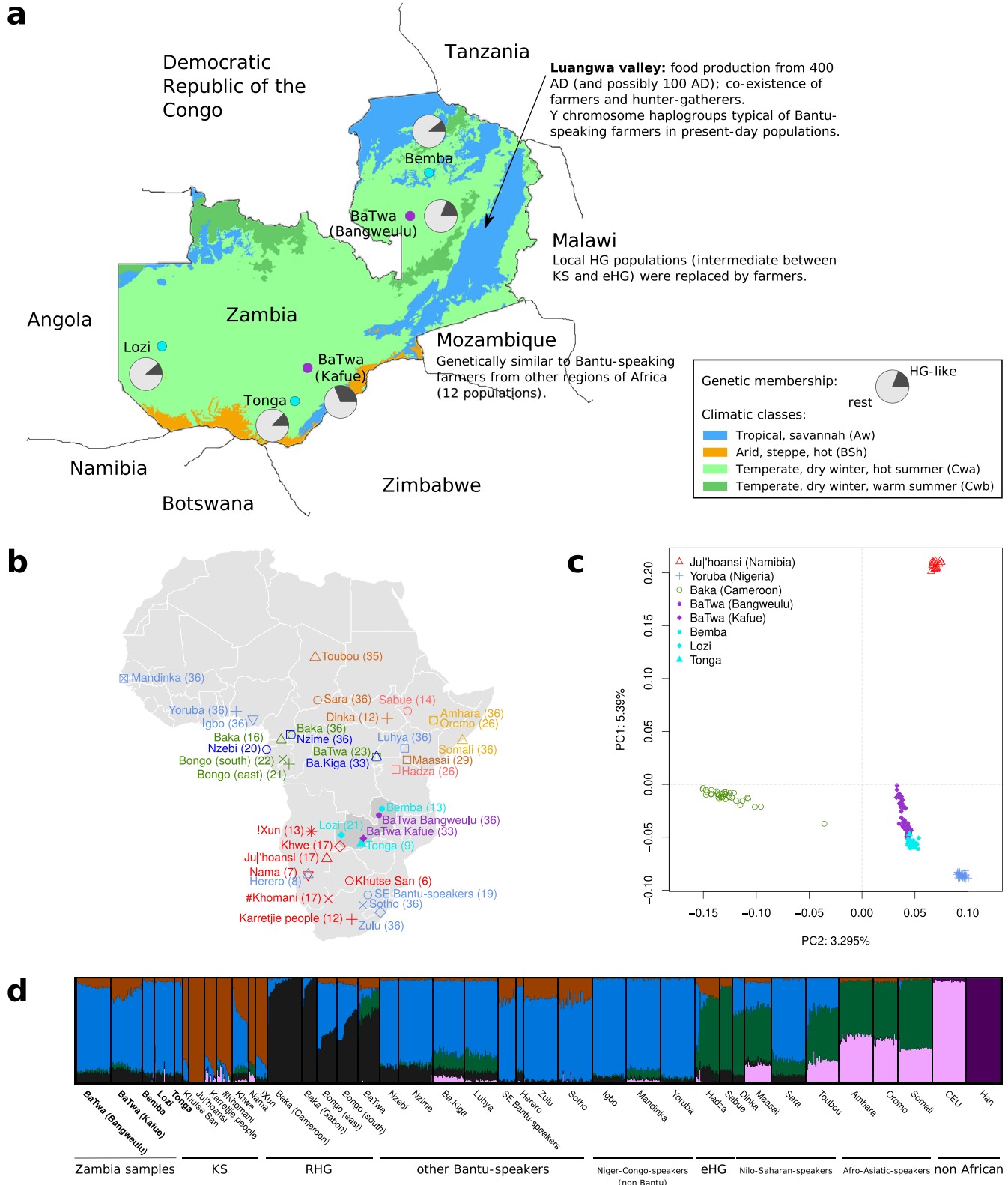

**Fig. 1 | Genetic affinities of five Zambian populations. a** Sampling locations of the Zambian populations, including information about the climatic classes, summarised evidence from other studies, and the proportions of the HG-like and non-HG like components based on unsupervised clustering (see Fig. 2 and Supplementary Table 2 for more details). The map of Zambia is modified from ref. 99 (version 1) according to the license CC BY 4.0 (https://creativecommons.org/licenses/by/4.0/legalcode). **b** Sampling locations of the Zambian and African comparative populations. Sample size (after removal of related individuals) is included in parenthesis after the population's name. **c** PCA (first two axes) of a subset of the Omni1 dataset (Yoruba, Ju|'hoansi and Baka) and projection of the five Zambian populations. See Supplementary Fig. 2 for an unprojected version. **d** Unsupervised clustering results for six putative clusters (see Supplementary Fig. 6 for the results with two to twelve clusters).

## Table 1 | Information about the Zambian populations

| Population | Sample size | Sampling location | Language | Lifestyle |
|---|---|---|---|---|
| BaTwa (Bangweulu) | 40 (36[a]) | Bangweulu wetlands, Luapula Province | Bemba | Fisherfolk |
| BaTwa (Kafue) | 40 (33[a]) | Kafue Flats, Central Province | Chitwa (dialect of Tonga or Ila) | Fisherfolk |
| Bemba | 13 | Kasama district, Northern Province | Bemba | Agriculturalist |
| Lozi | 21 | Mongu, Western Province | Lozi | Agropastoralist |
| Tonga | 10 (9[a]) | Southern Province | Tonga | Agropastoralist |

[a]Sample size after relatedness filtering.

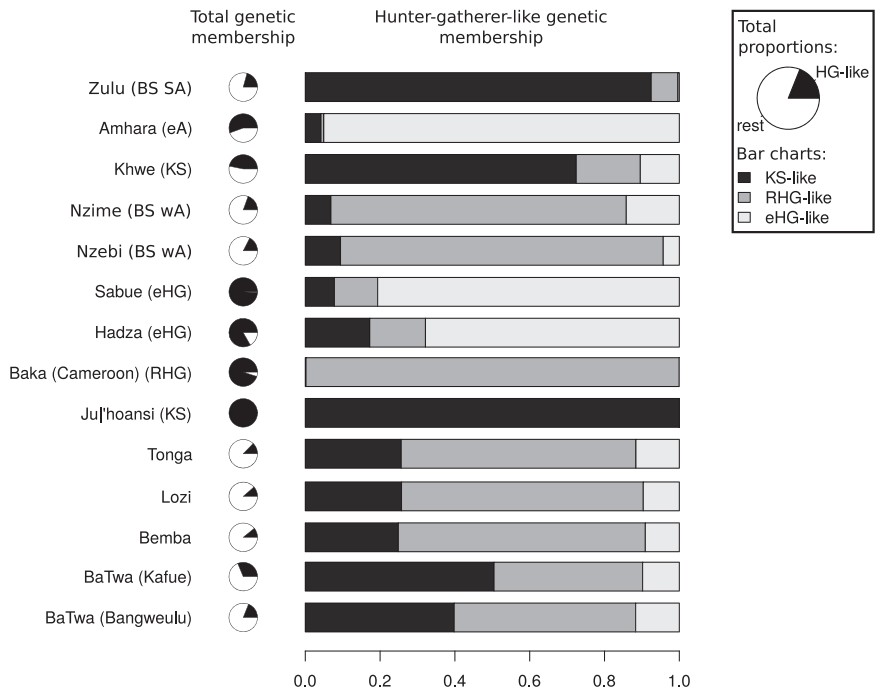

**Fig. 2 | Membership to genetic clusters for a subset of populations.** The memberships are based on the ADMIXTURE runs with six putative clusters. For each population, the pie chart represents the average total HG-like proportion (clusters maximised in Ju|'hoansi, Baka from Cameroon and Hadza) and the rest. The bar chart represents the relative proportions of the three HG-like proportions (population average of the individual ratios). The corresponding exact numbers are in Supplementary Table 2. BS SA: Bantu-speakers from South Africa; eA: eastern Africans; KS: Khoe-San; BS wA: Bantu-speakers from western Africa; eHG: eastern African hunter-gatherers; HG: hunter-gatherer.

agropastoralists may also have a contribution from a HG group, though to a lesser degree. Similar observations can be made with an unprojected PCA (Supplementary Fig. 2). If we replace the Yoruba by the Nzime, a Bantu-speaking population from Cameroon, which might be a better proxy for the ancestry of the agropastoralist migrants, we find similar results for the projected PCA (Supplementary Fig. 3). We also find similar results when we replace the western RHG Baka with the eastern RHG BaTwa from Uganda, which might be an alternative source for the HG genetic component (Supplementary Fig. 4). Further, similar patterns of relationships are observed if we use 39 populations to construct the PCA axes (Supplementary Fig. 1) or if we include data from ancient African individuals (Supplementary Fig. 5), suggesting that the choice of reference populations has little impact on the ancestry relations for the Zambian groups.

We inferred putative ancestry components using an approach implemented in ADMIXTURE[54], assuming between two and twelve genetic ancestry components (K) (Supplementary Fig. 6). At K=2, one cluster includes mainly individuals from African populations and the other includes mostly individuals from non-African populations. At K=3, a component maximised in the Ju|'hoansi (KS) appears. New components at K=6 are maximised respectively in the Baka from Cameroon (RHG) and in eastern African populations (representing

80.1% of the genetic membership of the Sabue -or Chabu- hunter-gatherer group from Ethiopia, standard error of mean (sem): 0.39%, Fig. 1d). The component maximised in the Niger-Congo speakers Yoruba (95.1%, sem: 0.23%) will be referred to as the western Africa-like ancestry in the following paragraphs. We note, however, that this does not reflect current-day population distributions and should be seen as an ancestral component; some of the populations that have a large proportion of this component, e.g., the Zulu, are Bantu-speakers from southern Africa that have ancestral connections to western Africa. Greater values of K's include components specific to single populations, e.g., Hadza from Tanzania (at K=7), BaTwa (eastern RHG) from Uganda (at K=9), and Sabue from Ethiopia (at K=10) and several of these populations are known to have small census sizes and thus to be more prone to genetic drift, which often shows up as unique ancestry components.

We focused on six putative ancestry components because there is a clear major mode at K=6, and it summarises the ancestry information for the Zambian individuals in a simple way. Four main components are present in the Zambian populations: a KS-like, a RHG-like, a western Africa-like (wA-like) and an eHG-like component. The two other components, maximised in the non-African populations, represent 0.01% to 0.31% in the Zambian populations, except for the Lozi, where it

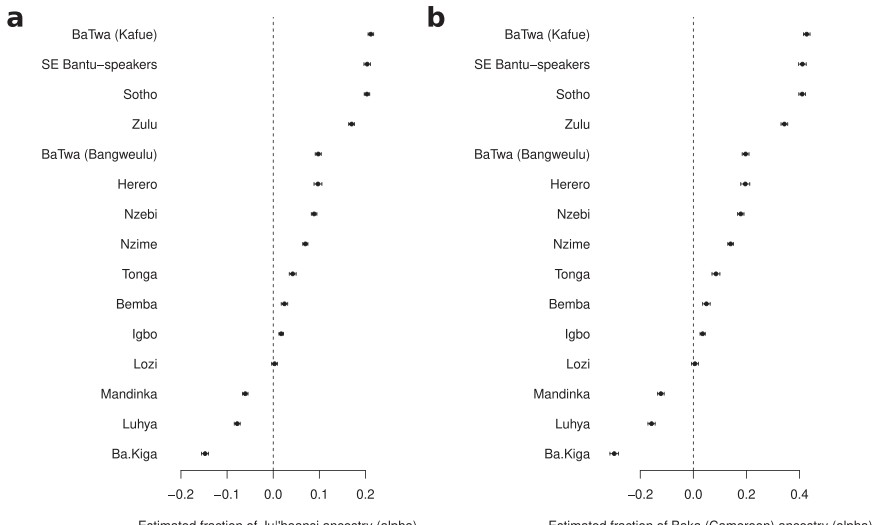

**Fig. 3 | Admixture proportions ($\alpha$) estimated with $f_4$ ratio test.** The test is of the form: $f_4$(Han Chinese, ancestral; target, Yoruba)/$f_4$(Han Chinese, ancestral; admixture source, Yoruba). The target populations are listed on the left. 337,051 SNP and 547 blocks for the block jacknife were used for the computation. The bars represent two standard deviations. **a** Admixture source: Khoe-San (Ju|'hoansi). **b** Admixture source: rainforest hunter-gatherer (Baka from Cameroon). The results are strongly correlated KS and RHG ancestries: Spearman rho correlation, two-sided, $p<2.2\times10^{-16}$, rho=1 (positive correlation). Negative $\alpha$ values can appear when the target population show no admixture from the admixture source and is genetically closely related to the comparative population (Yoruba).

represents 1.92% (this higher percentage is driven by two individuals). The sum of the KS-like, RHG-like, and eHG-like components represents ~31.3% of the genetic ancestry of the BaTwa from Kafue (sem: 1.4%), ~18.9% of the BaTwa from Bangweulu (sem: 0.5%), and 11–13% of the Zambian agropastoralists (Bemba: ~10.9%, sem: 0.6%; Lozi: ~10.5%, sem: 0.3%; Tonga: ~12.7%, sem: 0.6%) (Fig. 2, Supplementary Table 2). We calculated the relative proportions of the KS-like, RHG-like and eHG-like components in the non-western African component of the Zambian populations (Fig. 2, Supplementary Table 2). The relative proportion of the KS-like component is larger in the BaTwa populations (Kafue: 50%, Bangweulu: 40%) compared to the Zambian agropastoralists (~25%). The relative proportion of the RHG-like component is larger in the Zambian agropastoralist populations (~63–67%) than in the BaTwa (Bangweulu: ~49%, Kafue: ~40%). The eHG-like genetic proportion is lower than the KS-like and RHG-like components in the five Zambian populations, ~9–12%. Similar patterns of variable hunter-gatherer genetic affinities were observed when including data from ancient African individuals (Supplementary Figs. 7, 8).

## Characterising the nature of the admixture in Zambian populations

We tested whether the five Zambian populations were (two-way-) admixed between (i) a western African source (represented by the Yoruba) and (ii) a southern HG, central HG, or eastern HG source with $f_3$ statistics in order to determine the closest HG admixture-source population. Both pre-historic (including individuals from neighbouring Malawi) and present-day individuals were tested as potential source populations for the HGs. The BaTwa from Kafue show significant affinity to the Stone Age (2000 BP) individual from Ballito Bay ($f_3 = -0.003798$, Z-score $= -8.553$) and to the present-day San (Ju|'hoansi, $f_3 = 0.001653$, Z-score $= -7.802$), but not to the RHG populations (Supplementary Table 3). There appears also to be a genetic affinity to the pre-historic Malawi individuals, shown by low Z-scores (although they are non-significant, likely due to the limited amount of data from the pre-historic Malawi individuals). The four other Zambian populations show no clear affinity to any of the tested groups, possibly due to a combination of (i) a modest-low HG ancestry fraction, (ii) no perfect match of the HG ancestry component in the

Zambian populations to the tested source populations, and (iii) limited power of $f_3$ statistics.

In order to systematically explore multiple possible admixture events for the BaTwa from Kafue, we explored a large set of admixture-graphs using the `find_graphs` function of ADMIXTOOLS 2[55], with up to four admixture events. The best fit was a graph with three admixture events. For all tested combinations, only one admixture event involved the BaTwa from Kafue, which pointed to an admixture event between a HG population (in the graph represented by a group related to the Ju|'hoansi) and a group related to west Africans (Supplementary Fig. 9). Repeating the analysis for the other four Zambian populations that showed non-significant $f_3$-tests resulted in detection of a very small HG contribution (3% for Lozi, Supplementary Fig. 10) or no HG contribution as expected from the non-significant $f_3$-tests (Supplementary Figs. 11–13).

We further estimated admixture proportions from present-day HG populations using $f_4$ ratio tests. The $f_4$ ratio results (based on assumptions of population topologies) for the KS ancestry represented by the Ju|'hoansi (Fig. 3a, Supplementary Table 4) are consistent with the patterns suggested by the more model-free analyses (PCA, ADMIXTURE, Fig. 1c, d). The estimated ancestry proportion is ~21.2% (standard error: 0.6%) in the BaTwa from Kafue and ~9.7% (standard error: 0.6%) in the BaTwa from Bangweulu. For the Zambian agropastoralists the estimated proportions of KS admixture are ~4.2% in the Tonga, ~2.4% in the Bemba and ~0.3% in the Lozi. If we replace the potential KS admixture-source population with a RHG admixture-source population (represented by the Baka from Cameroon or the BaTwa from Uganda) the estimated HG admixture proportions are larger, but the same patterns appear: the estimated ancestry proportion is twice as large in the BaTwa from Kafue compared to the BaTwa from Bangweulu (Fig. 3b, Supplementary Fig. 14, Supplementary Tables 4, 5). However, we note that a single contribution from a RHG group to a west African group to create the Zambian populations was not supported by the $f_3$ tests (Supplementary Table 3). Finally, there is no evidence of ancestry from an eHG population (represented by the Hadza or the Sabue) into any of the Zambian populations (Supplementary Figs. 14, 15, Supplementary Tables 4, 5). The admixture fractions estimated for the different source populations are strongly correlated (positive correlation for the KS and RHG ancestries; the

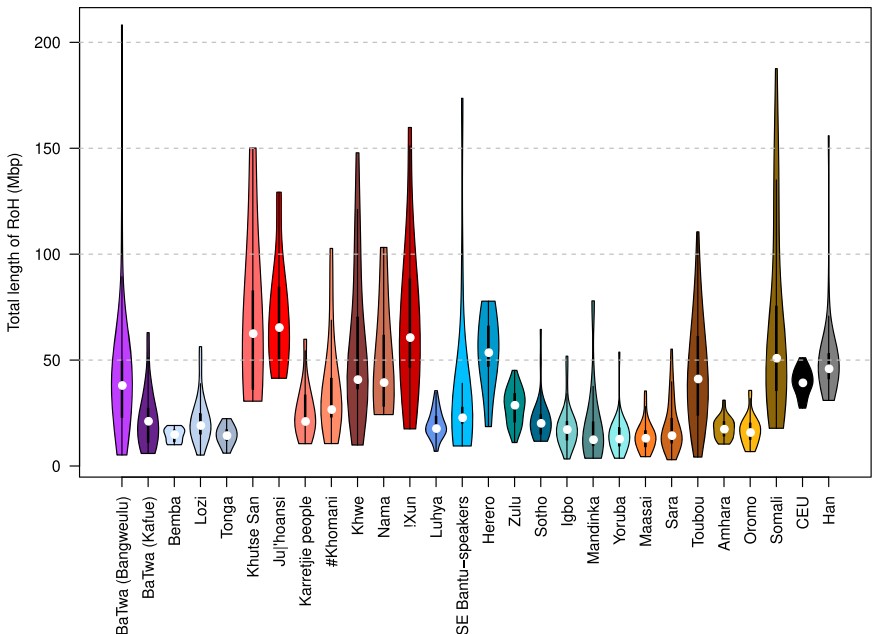

**Fig. 4 | Total runs of homozygosity length by individual by population.** The minimum length for a run to be included is 1 Mb. White dot, median. Wide black bar, lower quartile to upper quartile. The results subsetted according to class length are in Supplementary Fig. 18. Sample size from left to right: 36, 33, 13, 21, 9, 6, 17, 12, 17, 17, 7, 13, 36, 19, 8, 36, 36, 36, 36, 36, 29, 36, 35, 36, 26, 36, 36, 36.

western and eastern RHG ancestries; and the two eHG ancestries: Spearman rho correlation $p<2.2×10^{-16}$, rho=1.0; negative correlation for the KS and eHG ancestry, Spearman rho correlation $p<2.2×10^{-16}$, rho=−1.0). These trends show that the tests are detecting a similar signal of a non-west-African genetic ancestry component. In the absence of a matching proxy for this component, the tests appear equally suited to detect this non-west-African ancestry. The positive correlation between the KS and RHG ancestries fits with other observations of that the non-west-African component was intermediate between KS and RHG ancestries.

We used MOSAIC[56], a model-based approach, to characterise potential admixture events in the Zambian populations. A two-way admixture model with no specification of a reference panel (all the populations could be used to build the model and evaluated in terms of similarity with the admixing sources) is a good fit for the Zambian populations (Supplementary Figs. 16, 17 and GitHub repository[57]). The admixture event is estimated to ~38 generations ago for the BaTwa from Bangweulu (~1140 years ago with a generation time of 30 years) and to ~15 generations ago for the BaTwa from Kafue (~450 years ago). The time for the Bemba is ~23 generations ago (~690 years); it is older for the Lozi (~57 generations, ~1710 years) and the Tonga (~41 generations, ~1220 years). Bootstrap replicates and dates for additional populations are summarised in Supplementary Table 6.

For all five Zambian populations, the major ancestry component is estimated to be genetically most similar (smallest $F_{ST}$) to the Zambian agropastoralist populations, and the Nzebi and the Nzime (two Bantu-speaking populations from western central Africa). For the minor ancestry component, the closest potential source differs among Zambian populations (Supplementary Table 7). For the BaTwa from Kafue, the top five potential sources are modern-day KS populations. For the BaTwa from Bangweulu, the potential sources are the Khwe (a mixed population with KS and west African ancestry, see Fig. 1d), followed by: the BaTwa from Kafue; the Sotho, a South African Bantu-speaking population with admixture from KS; the Maasai from eastern Africa; and another KS population, the #Khomani. For the agropastoralists, the minor contributing source is closest to various eastern African populations. We note that the $F_{ST}$ values for the minor source

are distinctly larger for the BaTwa from Bangweulu and Kafue (~0.12 and ~0.06 respectively) than for the Zambian agropastoralists (~0.02) and for the five populations for the major source ($F_{ST}$ values in the order of 0.001). This suggests that our set of populations does not contain a population which is genetically close to the minor admixing source in the two BaTwa groups, in contrast to the agropastoralists who seem to have good population proxies also for the minor ancestry component. As a comparison, the minor source in the southeastern Bantu-speakers from South Africa is closest to various KS populations and the $F_{ST}$ values are ~0–0.02 for the five closest populations.

### Possible recent isolation in some Zambian populations

The distribution (lengths and counts) of runs of homozygosity (ROH) can indicate specific processes; for example long runs indicate small population size or inbreeding. The distribution of total ROH length (averaged by population) (Fig. 4) is similar to previous reports e.g., ref. 58. Two of the Zambian agropastoralist populations, the Bemba and the Tonga, are on the lower end of total ROH length, similar to other Niger-Congo-speaking populations such as the Mandinka and Yoruba. The Lozi and the BaTwa from Kafue have slightly larger total ROH length, while remaining on the lower end (median values: 19.5 and 21.5 Mb respectively, similar to the Karretjie people from South Africa: 21.4 Mb). The BaTwa from Bangweulu have the longest total ROH length from the Zambian populations (median: 38.4 Mb), and also the largest variance. However, the length is in the intermediate values, similar to populations such as the Khwe or the Toubou (median: 41.2 and 41.5 Mb respectively, large variance).

When the ROH are separated into length classes (Supplementary Fig. 18), the BaTwa from Bangweulu have the longest ROH length in the three classes among Zambian populations, while the BaTwa from Kafue and the Lozi have similar lengths to the Bemba and Tonga for classes 1–2 Mb and 2–3 Mb, and greater lengths for class 3–5 Mb.

The observation that the two BaTwa and the Lozi populations are elevated in the longest class compared to the Bemba and the Tonga, indicates recent isolation and inbreeding compared to the Bemba and the Tonga. This is more pronounced for the BaTwa from Bangweulu, which can also be seen in the ancestry components, where the BaTwa

population from Bangweulu forms its own ancestry component at $K$=12 (Supplementary Fig. 6). Geographical isolation might play a role in that both BaTwa communities live in seasonally flooded landscapes that are difficult to access, in particular for the Bangweulu group. However, the ROH distributions of the Zambian populations do not suggest extreme patterns that could impact the other analyses presented here.

## Paternal lineages in the Zambian populations

We identified the Y-chromosome haplogroup of the 74 Zambian male individuals with SNAPPY[59] (Supplementary Table 8, Supplementary Notes). In the BaTwa populations, we found a majority of haplogroups from the E-M2 sub-lineage, in particular E1b1a1a1a1c1a1a (E-U174), E1b1a1a1a2a1 (E-U209), and E1b1a1a1a2a1a3b1a (E-U290). The E-M2 sub-lineage is common among Bantu-speaking populations[35,60] and the specific haplogroups that we identified were reported previously in other Zambian populations[35,37] (Supplementary Tables 9, 10). We also identified six copies from the major haplogroup B, which shows a differential distribution among the Zambian populations: B2a in the agropastoralists and B2b in the BaTwa. This might reflect a difference in ancestry between agropastoralists and BaTwa since the sub-lineage B2b is associated with African hunter-gatherer populations[61,62]. Overall however, we found a majority of typical Bantu-speaker paternal lineages in the BaTwa.

The maternal lineages of some Zambian groups have previously been investigated[37], and the H3Africa v1 callset was not informative enough for calling mitochondrial haplogroups for African populations[63].

## Sex-biased admixture

Contrasting the X chromosome and the autosomes in terms of ancestry is informative about sex-biased admixture[64–67]. We calculated the female and male contributions from a western-African source and a HG source (represented by the Yoruba and Ju|'hoansi, respectively) and the corresponding sex ratios in a subset of populations, including the Zambian BaTwa and agropastoralists, KS, RHG and RHG neighbour populations (Supplementary Table 11). We also calculated the X chromosomal to autosomal ancestry ratio (Supplementary Table 12). The ratios of the female to male western-African contributions are less than 1 in the Zambian populations, at -0.79–0.83 in the agropastoralists and -0.58–0.70 in the BaTwa, indicating a male biased contribution of the western-African-like ancestry (Supplementary Table 11). The ratios for the hunter-gatherer ancestry are negative in the Zambian populations, indicating model violation. Their absolute values are greater than 1, which combined with X chromosomal to autosomal ancestry ratios greater than 1 (Supplementary Table 12), suggests a female biased contribution of the HG source.

Overall, these results and the mostly typical Bantu-speaker Y chromosome haplogroups indicate that the admixture that resulted in today's Zambian populations was male-biased for the western-African-like ancestry, which is in line with previous results describing the Bantu expansion[68–74].

## Discussion

Chronologically, Bantu-speakers represent the most recent population layer in sub-equatorial Africa. Bantu-speakers either replaced or further fragmented the complex population mosaics that existed in earlier time periods. In some areas of southern, central and eastern Africa, groups that are likely the descendants of the earlier population strata still remain, such as the Khoe-San, rainforest hunter-gatherers, and Hadza. While pre-existing groups in other regions were primarily absorbed and/or replaced by Bantu-speaking groups, there are indications of their prior presence from ancient DNA studies (e.g., in Malawi[17]) and archaeological surveys (e.g., in the Luangwa valley[26]). It is crucial to bridge gaps in our understanding of these pre-Bantu-speaking groups, a task that can sometimes be achieved by directly

investigating the genetic information in human remains from relevant time periods, but for many regions and time-periods, human remains that preserved DNA have not yet been found. The alternative is then to investigate the potential descendants of the earlier population strata that often mixed with Bantu-speakers. For example, groups intermediate between KS and RHG proxies and groups that seem to represent an unknown ancestry of pre-Bantu-speaker foragers were identified in Angola (southwestern Africa)[22].

Similarly, we investigate genome-wide data for two BaTwa populations from Zambia who live as fishers and foragers: the BaTwa from the Kafue Flats and the BaTwa from the Bangweulu wetlands, together with three Zambian agropastoralist populations and comparative data representing African and non-African diversity. The analyses provide a genetic-based confirmation that while these BaTwa communities are genetically largely similar to other Zambian populations, they carry a distinct minor genetic component (31% in the Kafue and 19% in the Bangweulu) that likely originate from a local and distinct HG population, with distant affinities to present-day HGs (intermediate between KS- and RHG-populations).

The major ancestry component for the BaTwa is closest to the Zambian agropastoralist Bantu-speaking populations and to western-central Bantu-speakers from Cameroon and Gabon. The admixture with the minor HG source is dated to ~450 years ago for the BaTwa from Kafue, and more than twice as old at ~1140 years for the BaTwa from Bangweulu. Archaeological evidence in the form of calibrated radiocarbon dates for the earliest Iron Age sites in the immediate vicinity of each present-day BaTwa population provide support to these admixture-date estimates. For the Kafue locality, the Iron Age site of Sebanzi Hill[75] places agropastoralists on the edge of the flood-plain ~1525 cal AD (98.9% probability) (750 ± 135 bp, GX 109, charcoal[76], SHCal 13, range: 935 cal AD (0.2%), 1525 cal AD (98.9%), 1627 cal AD (0.6%)). The initial settlement of Sebanzi Hill may be a century earlier based on the depth of the undated underlying sediment[75], but this estimate does not affect the close correspondence between the genetic admixture calculation and the archaeological evidence. For the Bangweulu locality, archaeological evidence from the site of Samfya, which is within BaTwa area, indicates an arrival of farmers ~682 cal AD (94.5% probability) (1520 ± 80 bp, N-1934, charcoal[77], SHCal 13, range: 398 cal AD (0.3%), 682 cal AD (94.5%), 757 cal AD (0.6%)) which accords with an earlier admixture time frame in this part of Zambia.

Overall, the estimated dates are compatible with the western-African-like component being brought to Zambia during the Bantu expansion[2]. The estimated date for the admixture in the BaTwa from Bangweulu is similar to the estimated dates of admixture in southern Mozambican and South African Sotho and Zulu populations (between a major Bantu-speaker farmer ancestry and a minor KS ancestry)[38]. Looking at languages, the BaTwa from Kafue speak a dialect of the dominant local language Tonga. The dialect, Chitwa[50], reflects their isolation from mainland Tonga and resistance to adopting their lifestyle. The BaTwa from Bangweulu have a longer history of contact with agriculturalists and presumably have lost their linguistic identity through assimilation.

The minor ancestry component is best described as intermediate between a KS- and a RHG-like component, with slightly greater KS affinity (or fraction), especially in the population from Kafue (Figs. 1, 3, Supplementary Table 3). It could have originated from local Zambian HG populations (of which we do not have a representation); or it could be the result of recent migration and admixture from KS groups from southern Africa. However, present-day KS populations are not a good match for the minor component (Supplementary Table 7). In particular, the closest minor source for the BaTwa from Bangweulu is the Khwe population, whose genetic makeup is a mixture of KS and west African ancestry[12]. The other potential sources of the minor component in the BaTwa from Bangweulu are very diverse and genetically different. This suggests that the source could rather have been a local

population in Zambia, distantly related to present-day southern African KS groups which accords with the current interpretation of Middle and Late Holocene HG skeletal evidence from Zambia and Malawi. The morphological examination of the cranial remains from the 4000-year-old Later Stone Age site of Gwisho A on the margins of the Kafue Flats[23] does not show a exclusive similarity to KS groups[24]. A similar finding is reported for Middle and Late Holocene crania (5000 to 500 years ago) from Malawi which do not match any extant HG population[25]. HGs of southcentral Africa appear to have been morphologically distinctive, supporting the interpretation of the genetic results. Interestingly, the BaTwa do not present an eHG genetic component, in contrast to prehistoric individuals from Malawi or an individual who lived 5000 years ago in current-day Zambia, some 500–700 km away from the BaTwa populations[21]. The ancestry cline (between eastern African and southern African ancestries) among these ancient individuals has been interpreted as a migration corridor during the Middle Holocene[17], but it appears as if this gene-flow did not extend into the central plateau of Zambia, and it might have been a more regional gene-flow to Malawi from eastern Africa. The BaTwa of Zambia do not present a pastoralist associated eastern African component (Fig. 1d, Supplementary Figs. 1, 5, 6, 7, 8), and as expected, the fraction of lactase-persistence variants is very low (for the two variants on the chip: 1 C- and 153 G-alleles at position -14,010 in the LCT gene: 0.6%; no -13,907 G-allele). Hence, the eastern African pastoralist migrants bringing pastoralist practices to southern Africa[16,78] did not reach the western parts of (today's) Zambia to admix with the BaTwa of Zambia.

The HG component in the Zambian groups has an RHG affinity (Figs. 1, 2) in addition to a KS affinity. The BaTwa populations appear to have a stronger RHG affinity (or a greater fraction) than the neighbouring agropastoralist populations (Supplementary Notes, Supplementary Figs. 19–23), which suggests that a more RHG-like HG group contributed the minor source to the ancestors of the BaTwa. Another scenario would be that the major source of the BaTwa was contributed by a first wave of farmers coming to Zambia, who carried a greater fraction of RHG ancestry, while the ancestors of the Bemba, Lozi and Tonga arrived via a later migration of Bantu-speaking farmers with less RHG ancestry.

The BaTwa mostly have typical Bantu-speaking farmers Y-haplogroups; thus with Y chromosome data only, we would not have been able to identify the HG-like component in the BaTwa. This demonstrates the need for autosomal data to identify potential pre-Bantu ancestry. Local Y lineages were often replaced by incoming populations during the Bantu expansion[68–73]. We found a similar pattern in the BaTwa from Zambia; our results support male-biased contribution from the western-African-like source into the BaTwa, while values for the HG-like source suggest a female-biased contribution (Supplementary Tables 11, 12). Model failure might be explained by the lack of an appropriate proxy for the HG-like-source. These results are consistent with ref. 37, who proposed female-biased admixture from Khoisan speaking groups into a Bantu-speaking population to explain the patterns of Y chromosomal and mitochondrial diversity in two populations from Zambia, the Fwe and the Shanjo. The ancestors of these two populations lived initially in the Kafue plains according to linguistic evidence, before migrating to the southwest[79]; this could suggest a shared history with the BaTwa from Kafue. Archaeological evidence also indicate a 2nd millennium AD presence of HGs along the mid to upper Zambesi that overlaps with the geographical range of the Fwe[80,81], but again further genetic research is needed to address issues of population affinities raised by the archaeological record.

There are several differences between the two investigated BaTwa populations, such as the proportion of HG-like ancestry; the inferred admixture time; and the populations that are the best proxies for the minor admixture source. This raises the question of whether the local population source was different between the two localities.

The agropastoralists Bemba, Lozi and Tonga also have a small HG-like genetic component, corresponding to 11–13% in the ADMIXTURE analysis; 25% of this is KS-like, and 65% is RHG-like (this is less KS-like, and more RHG-like, than the BaTwa) (Fig. 2, Supplementary Table 2). The MOSAIC analysis showed that a simple two-way admixture model is a good fit for the Zambian agropastoralist populations, with a major source closest to other Zambian agropastoralist populations or to the Nzime and Nzebi. Admixture events inferred in the Zambian agropastoralists involve an eastern African population (non-HG) as minor source (for example Oromo, Maasai or Amhara), similar to observations from current day Bantu-speaking farmer populations from the Great Lakes, coastal Kenya and Tanzania[38] as well as from prehistoric individuals from eastern Africa[17]. The $F_{ST}$ estimates between the minor source and its closest proxy in our dataset are lower than in the BaTwa (Supplementary Table 7), suggesting that the minor admixture sources in the agropastoralists are closer to populations in our dataset. The estimated admixture times differed for the three Zambian agropastoralist populations, and are either on the older end (Lozi and Tonga, 57 and 41 generations ago respectively) or on the younger end (Bemba, 23 generations ago) of the dates estimated elsewhere for Bantu-speaking populations from eastern Africa[17,38]. In Zambia, the Tonga could represent an early arrival of Bantu-speakers based on comparative linguistic data[79]. Taken together, these observations suggest that the HG-like genetic component in the Zambian agropastoralists has a different origin than in the BaTwa.

Genome-wide autosomal data from two BaTwa populations from Zambia revealed that they harbour a HG-like genetic component, representing ~19% and ~31% of their genetic makeup respectively. This component likely originates from local HG populations, distantly related to KS from southern Africa and to RHG from central Africa. This is the first direct genetic evidence that the BaTwa trace a portion of their ancestry to HG populations that predate the arrival of Bantu-speaking farmers. Admixture dates between the autochthonous HG groups and incoming Bantu-speaking farmers is estimated to have happened ~38 generations ago for the northern and ~15 generations ago for the central population. These results highlight differences in population histories from neighbouring countries such as Mozambique and Malawi, where local HG genetic components are close to non-existent. The integration of our genetic results with Middle Holocene skeletal data from Zambia and Malawi adds support to the hypothesis of a separate southcentral African HG population that once existed in the region. The retention of a greater HG related ancestry-component among the BaTwa compared to the agropastoralists may reflect the geographical isolation of these communities in seasonally flooded wetlands. These results underline the value of studying underrepresented groups such as the BaTwa for reconstructing the complexity of human population history in Africa, in particular the history of the people that lived throughout Africa prior to the arrival of agropastoralists from western and eastern Africa in the last few millennia.

## Methods

### Sampling and genome-wide SNP typing

Individuals were sampled in 2017 from two locations in Zambia where BaTwa live: (i) Lake Bangweulu (around Bwalya Mponda, located at 11°40'0.41" S; 30°04'28.02" E), and (ii) Kafue Flats (located at 15°53'30.51" S; 27°13'20.93" E). George Mudenda and Perrice Nkombwe (Livingstone Museum, Zambia) sampled 40 individuals from each location. Ethical clearance was granted by the Committee on Research Ethics of the University of Liverpool, UK (permit number RETH001037), as well as the University of Zambia Biomedical Research Ethics Committee (permit number 016-07-15), the National Health Research Authority, Zambia (permit number MH/101/23/10/1) and the Swedish Ethical Review Authority (Dnr 2021-04013). Written

informed consent was obtained from the participants before saliva samples were collected. In 2022, Lawrence Barham, Gwenna Breton and George Mudenda returned to the two locations to present the results to the participant groups and discuss the results with them. The results were well received and the people showed great interest in the findings.

The BaTwa are politically marginalised ethnic communities and socially stigmatised by their Bantu-speaking farmer neighbours[40,49]. The stigma attached to the name BaTwa was reflected in the sampling design. Since individuals may have been reluctant to self-identify as BaTwa, they were not asked to do so directly. They were instead asked for their clan affiliation and that of their parents and grandparents. Clan membership can be an indicator of BaTwa affinity[48]. Moreover, participants whose families have been resident locally over three generations were asked to participate in the study to improve the chances of sampling those with BaTwa ancestry. Information about the populations can be found in Table 1 and Supplementary Table 1; the sampling locations are indicated in Fig. 1a.

DNA was extracted from the saliva samples from BaTwa populations with the prepIT DNA isolation kit (DNAgenotek Inc., Ottawa, ON, Canada) according to kit instructions.

Genome-wide SNP typing on the H3Africa Consortium Array v1 (H3Africa_2017_20021485_A2)[52], implemented on the Illumina Infinium array, was performed at the SNP&SEQ Technology Platform in Uppsala, Sweden. The results were analysed with the software GenomeStudio 2011.1 from Illumina Inc, and aligned to the human genome GRCh37. While the H3Africa array was designed to limit ascertainment bias in African populations, we note that arrays are ascertained by definition[82]. Moreover, the data was merged with other datasets typed on other arrays (see below), which weren't designed to limit bias in African populations.

### Quality filtering and autosomal dataset merging

The Zambian BaTwa dataset was quality filtered. It was then merged with a (quality filtered) Zambian agropastoralist dataset, also typed on the H3Africa array. Further merging with comparative datasets typed either on the Illumina Omni1 or Omni2.5 array was performed, as well as with ancient individuals. Finally, the merged present-day human datasets were phased. Details of the quality filtering, merging and phasing are as follows.

**Quality filtering (Zambian BaTwa dataset).** The quality filtering and merging was performed with plink[83] v1.90b4.9. The initial dataset contained 2,267,346 variants. A fake individual, heterozygous at every position for the two alleles typed on the array, was added to the fileset to avoid loss of information at the merging steps. SNPs located on the sex chromosomes and the mitochondria, or not attributed to a specific chromosome, as well as all indels, were excluded. The fileset was filtered for variant missingness with a threshold of 0.1 (`-geno 0.1`) and for individual missingness with a threshold of 0.1 (`-mind 0.1`). A fileset was created for each population in order to identify variants with a deviation from Hardy-Weinberg equilibrium (HWE) with a $p \leq 0.001$ (`-hwe 0.001`). Variants deviating from HWE in both BaTwa populations were excluded (205 variants). Variants names were changed to the format `chr:position`. Duplicate variants were identified and excluded. Relatedness was investigated with two methods: KING v2.1.4[84] and plink `-genome`. Pairs of first and second degree relatives (defined in KING as pairs of individuals with a kinship value superior to 0.0884) were examined in order to remove related individuals while maximising the number of individuals; in a given pair, the individual with the highest missingness was excluded. A-T and G-C variants were excluded to prevent strand errors while flipping during the merging process.

After quality filtering, 69 individuals (33 from Kafue, 36 from Bangweulu) remained in the BaTwa dataset.

## Merging

**Zambian agropastoralist dataset.** Three Bantu-speaking agropastoralist populations from Zambia were included as comparative neighbouring groups (Table 1, Supplementary Table 1): the Bemba ($n = 13$, northeast, neighbours of the BaTwa from Bangweulu), the Lozi ($n = 21$, northwest, prestige group in western Zambia), and the Tonga ($n = 10$, southwest)[53]. We refer to these groups as Zambian agropastoralists to differentiate them from the BaTwa groups. The Tonga and Lozi are cattle-keepers.

The DNA samples were typed on the H3Africa array. The quality filtering and merging was performed with plink[83] v1.90b4.9. A fake individual, heterozygous at every position for the two alleles typed on the array, was added to the fileset to avoid loss of information at the merging steps. SNPs located on the sex chromosomes and the mitochondria, or not attributed to a specific chromosome, as well as all indels, were excluded. The fileset was filtered for variant missingness with a threshold of 0.1 (`-geno 0.1`) and for individual missingness with a threshold of 0.1 (`-mind 0.1`). A fileset was created for each population in order to identify variants with a deviation from Hardy-Weinberg equilibrium (HWE) with a $p \leq 0.001$ (`-hwe 0.001`). Variants deviating from HWE in at least two of the three agropastoralist populations (4 variants), were excluded. Variants names were changed to the format `chr:position`. Duplicate variants were identified and excluded. Relatedness was investigated with two methods: KING v2.1.4[84] and plink `-genome`. Pairs of first and second degree relatives (defined in KING as pairs of individuals with a kinship value superior to 0.0884) were examined in order to remove related individuals while maximising the number of individuals; in a given pair, the individual with the highest missingness was excluded. A-T and G-C variants were excluded to prevent strand errors while flipping during the merging process. Following visual inspection of the data via multidimensional scaling of the allele-sharing-dissimilarity matrices[85] (calculated with the software asd v1.0[86]), an individual from a pair of more than average related individuals in the Bemba population was also removed (as that pair was driving one of the MDS first three axes).

After quality filtering, 43 individuals (13 Bemba, 21 Lozi, 9 Tonga) remained in the Zambian agropastoralist dataset.

The BaTwa and the agropastoralist datasets were merged with plink. The merging was performed step-wise, and included a fake individual (heterozygous at every position). In case of conflicts between filesets, the strand orientation was flipped; remaining mismatches after strand flipping were excluded. A variant missingness filter of 0.1 was applied after each merge to retain intersecting variants.

**Present-day human datasets.** Comparative datasets were selected to cover the main ancestries in Africa and in the world[12,13,87–90]. In particular, we included Khoe-San (KS), central African rainforest hunter-gatherers (RHG) and eastern African hunter-gatherers (eHG) populations. Preference was given to datasets typed on the Illumina Omni1 or the Illumina Omni2.5 arrays. Details of the included populations are given in Supplementary Table 1 and sampling locations are shown in Fig. 1b. Prior to merging, datasets were filtered in a similar way than for the Zambian datasets: HWE filtering (overlap of variants deviating from HWE in two populations, even if there were more than two populations); relatedness filtering; variant and individual missingness; and exclusion of A-T and C-G sites. Population samples were randomly downsampled to 36 individuals to ensure equivalent sample sizes. We note that there are two Baka populations[87]; they are distinguished by their sampling location (respectively Cameroon and Gabon). For analyses involving a single Baka population, we chose the population from Cameroon. There are also two Bongo populations from Gabon, distinguished as east and south[87].

The merging was performed step-wise, always including a fake individual (heterozygous at every position). We merged the Zambian dataset first with the individuals from[12]; then with[87–90] and[13]. In case of

conflicts between filesets, the strand orientation was flipped; remaining mismatches after strand flipping were excluded. A variant missingness filter of 0.1 was applied after each merge to retain intersecting variants; for the last merge of the Omni1 fileset, the threshold was set to 0.05. The Omni2.5 fileset contained 1,187,130 variants and 714 individuals; the Omni1 fileset contained 344,644 variants and 973 individuals.

**Present-day and ancient human dataset.** We obtained the plink fileset of ancient African individuals presented in ref. 19 from which we excluded the individuals from Morocco and Egypt. The data from the remaining individuals were initially published in refs. 15–18,91 (Supplementary Table 13). The dataset contains shot-gun sequences (whole genomes) as well as capture data. The ancient individuals were merged with the Omni1 dataset. The resulting dataset was haplodized (i.e., for the diploid individuals, an allele was picked at random for every bi-allelic site) using a custom script (`haploidize_tped.py` in the GitHub repository[57]), and filtered for individual and site missingness to ensure that each individual had at least 15,000 non-missing genotypes (plink `-geno 0.15 -mind 0.956`). The final dataset had 344,644 variants and 1,019 individuals. This dataset was used for population structure analyses and for calculation of $f_3$ statistics.

### Phasing
The Omni1 dataset was phased with SHAPEIT2 v2.r904[92] with the 1000 Genomes phase 3 haplotype panel[88] and the 1000 Genomes combined map for hg37. Prior to phasing, SHAPEIT2 `check` command was used to align between the target dataset and the reference panel. Variants with strand inconsistencies between the two panels were flipped in the target dataset. The variants which were still inconsistent after the allele flipping were excluded. Graphs were obtained with SHAPEIT2 `-output-graph`, and were converted into .haps filesets with SHAPEIT2 `convert -output-max`.

### X chromosome and autosomes fileset
We assembled a different dataset comprising the autosomes and the X chromosome for the populations for which we had X chromosome data, i.e., the five Zambian populations and comparative data from[12,87,88]. We proceeded in a similar way than when we assembled the autosomes-only datasets, though the order of some steps was changed; we removed the same individuals due to relatedness. We note that we did not start from exactly the same files for the data in[12,87]. After filtering for linkage disequilibrium (LD) (`-indep-pairwise 50 10 0.4`), 382,167 variants remained of which 3729 on the X chromosome. This dataset was used to compute the X chromosome to autosomes ancestry ratio.

### Population structure analyses
Descriptive analyses were performed on various datasets and subsets of populations to visualise the diversity in the data.

**Principal component analysis.** PCA was performed with eigensoft v7.2.0 smartpca[93], to identify principal components that explain most of the variance in the dataset. For the present-day data, the maximum population size was set to 36 and the `killr2` option was used, with a r2 threshold of 0.2. Default values were used for the other parameters. PCA with projected individuals were also performed using the `poplistname` option. PCA results were plotted with R v3.6.3[94] (Fig. 1c, Supplementary Figs. 1–4).

We performed a PCA on the dataset including ancient individuals (Supplementary Fig. 5). We restricted the analysis to African individuals only. An ancient individual, the high coverage ancient southern African from Ballito Bay (baa001 in[16]) was included in the population reference list along with the present-day individuals. The rest of the

ancient individuals was projected. We used eigensoft v.7.2.0 smartpca, with options `lsqproject (YES)`, `shrinkmode (YES)`, `killr2 (YES)`, a `r2` threshold of 0.2, and `altnormstyle (NO)`.

**ADMIXTURE.** ADMIXTURE v1.3.0[54], an unsupervised clustering approach which identifies blocks of genetic diversity which are shared between different individuals, was applied to the Omni1 dataset. The dataset was first pruned for LD, using plink `-indep-pairwise 50 10 0.1`, resulting in 115,257 variants. We tested two to twelve putative genetic cluster (K), with twenty repeats for each K and a random seed. This was considered sufficient as most of the K clusters were resolved. ADMIXTURE results were plotted with pong[95], which identifies the major mode for each of the K clusters. A single mode was found for 2, 3, 4, 6, 7, 10 and 11 putative genetic clusters. The cross-validation error is shown in Supplementary Fig. 24.

The dataset including ancient individuals was pruned for LD with the same parameters, resulting in 243,633 variants in 1019 individuals. We ran 20 repeats for values of K from two to twelve. When relevant, we grouped ancient individuals into families (e.g., the two Shum Laka individuals dated to -3000 BP[18]). Note that the population sample sizes are very variable, with small sample sizes (as low as one individual) for the ancient individuals. To ease visualisation, we chose the representative run of the most informative K (K = 6) and calculated the average of the genetic membership for the present-day populations (Supplementary Fig. 7). Thus, the sample size of present-day populations appears to be 1, but it represents a population average and not individual proportions. The full ADMIXTURE plot is shown in Supplementary Fig. 8.

### Statistical testing (admixture proportions)
We compared genetic membership proportions for different populations, using the representative run of the major mode at K = 6 for the ADMIXTURE with the Omni1 dataset (output of pong with the verbose option) (Supplementary Fig. 6). In particular, we calculated the ratio of the RHG-like component relative to the western-African like component for the individuals in a subset of populations (Supplementary Fig. 19). We tested whether the BaTwa from Kafue (respectively from Bangweulu) had a greater mean (i.e., larger proportion of RHG-like component) than the Zambian agropastoralists. We used R t.test one-sided function (Welch two-sample t-test)[94]. Similarly, we tested the equality of the mean between the southern African Bantu-speakers (Herero, Sotho, Zulu, and southeastern Bantu-speakers) and the Lozi; between the three pairs of Zambian agropastoralist populations; and between the two BaTwa populations, using a one- or two-sided test depending on the comparison.

### Test admixture hypotheses with $f$-statistics
**Test admixture hypotheses with ancient and present-day sources with $f_3$ statistics.** The $f_3$ statistics were calculated with AdmixTools v5.0-20171024 qp3Pop program[96]. The input plink fileset was converted to eigenstrat format (.indiv, .geno and .snp files) with `convertf`. We used the option `inbreed: yes` since we are working with pseudo-haploid data. In each test, the target population was one of the five Zambian populations. We tested whether there was evidence that the target population was admixed between western Africans (represented by the Yoruba) on one hand, and one of the following populations on the other hand:

1. Rainforest hunter-gatherers, represented either by the Baka from Cameroon (present-day), or the ancient individuals from Shum Laka, Cameroon, grouped according to their age (3000 BP or 8000 BP)[18].
2. San, represented either by the Ju|'hoansi from Namibia (present-day), or the late Stone Age individual from Ballito Bay who lived 2000 years ago (baa001)[16].

3. Eastern African hunter-gatherers, represented either by the Hadza from Tanzania (present-day), or three ancient individuals from Malawi, that span several thousand years but are genetically homogeneous: Malawi_Fingira_6100BP (I4427, radiocarbon dated to ~6000 BP), Malawi_Fingira_2500BP (I4426, radiocarbon dated to ~2500 BP), and Malawi_Hora_8100BP Holocene (I2966, indirectly dated to ~10,000–5000 BP)[17].

**Explore admixture graph.** We explored admixture graphs with ADMIXTOOLS 2 `find_graphs` function, using the admixtools v2.0.0 R package[55]. We included a Denisovan genome as outgroup. The genome was published originally in ref. [97] and remapped in ref. [98]. The remapped BAMs[98] were obtained and genotypes were called in ref. [16]. Here, we extracted the positions in the Omni1 dataset from the Denisovan genome all-sites VCF. Then we followed our standard merging procedure (rename variants, exclude ATCG, flip alleles, remove inconsistent variants). This resulted in a set of 337,051 variants.

We calculated all $f_2$ statistics from a plink fileset with: `f2_from_geno("fileset", maxmiss=0.15)`. We then ran `find_graphs` as follows: `find_graphs(f2, numadmix = m, stop_gen = 10000, stop_gen2 = 30, plusminus_generations = 10, outpop = 'Denisovan')`. We tested three values of $m$, the number of admixture events: one, two or three. For the BaTwa from Kafue, we also tested four admixture events. We ran 100 iterations for each $m$. We then ran `qpgraph_resample_multi` on the graph with the lowest score for each $m$, with 1,000 bootstraps. We compared the fit of models with one or two admixture events, with two or three admixture events, and with three or four admixture events, with `compare_fits`.

We analysed each of the five Zambian populations with five other populations: Ju|'hoansi, Baka (Cameroon), Hadza, Yoruba, and Denisova. Two Lozi individuals with recent non-African admixture were excluded from the analyses. For the BaTwa from Kafue and the Lozi, three admixture events were a significantly better fit than two admixture events (BaTwa from Kafue: $p_{emp} = 0.014$, Lozi: $p_{emp} = 0.002$). For the other three sets, three admixture events were not a significantly better fit than two; two events were a significantly better fit than one (all: $p_{emp} = 0.001$). For the BaTwa from Kafue, four admixture events were not a significantly better fit than three ($p_{emp} = 0.87$).

**Quantify admixture with $f_4$ test ratios.** $f_4$ test ratios were calculated with AdmixTools v5.0-20171024 qpF4ratio program[96], to estimate ancestry proportions in admixed populations. We merged the Omni1 dataset with a plink fileset containing ancestral state information for the H3Africa callset. The ancestral state was inferred using an alignment of human and three great apes: chimpanzee (panTro4, UCSC, downloaded from ftp://hgdownload.soe.ucsc.edu/goldenPath/panTro4/), gorilla (gorGor3, UCSC, ftp://hgdownload.soe.ucsc.edu/goldenPath/gorGor3/), and orangutan (ponAbe2, UCSC, ftp://hgdownload.soe.ucsc.edu/goldenPath/ponAbe2/). For a site to be included, the three non-human great apes needed to have the same allele. We calculated $f_4$ ratios in the following way: $f_4$(unrelated, ancestral; target, main ancestry proxy)/$f_4$(unrelated, ancestral; admixture source, main ancestry proxy). The following was the same in all the tests: the Han Chinese were the unrelated population; the three non-human apes were the ancestral population; the Yoruba were the main ancestry proxy (representing western African ancestry). We tested five admixture sources: the Ju|'hoansi, the Baka from Cameroon, the BaTwa from Uganda, the Sabue, and the Hadza, representing respectively the KS, the western and eastern RHG and the eHG. For each admixture source, we tested the different Niger-Congo speaking populations (including the BaTwa) as target. As an example, for testing admixture from KS into the BaTwa from Bangweulu, the test was: $f_4$(Han Chinese, three apes; BaTwa (Bangweulu), Ju|'hoansi)/$f_4$(Han Chinese, three apes; Yoruba, Ju|'hoansi). Results are presented in Fig. 3 and Supplementary Figs. 14, 15.

**Runs of homozygosity**

Runs of homozygosity (ROH), i.e., stretches in each individual genome where the two alleles are identical (homozygous), were identified. They are informative in terms of recent inbreeding or bottlenecks. ROH were characterised in the denser Omni2.5 dataset. Prior to characterisation, the dataset was filtered more stringently for individual and variant missingness, with a threshold of 0.05 (similar to ref. [58]), resulting in a dataset of 1,176,055 variants and 714 individuals.

ROH were identified with plink `–homozyg –homozyg-snp 50 –homozyg-kb 1000 –homozyg-density 50 –homozyg-gap 100 –homozyg-window-snp 50 –homozyg-window-het 1 –homozyg-window-missing 5 –homozyg-window-threshold 0.05`[83]. Information extracted from the outputs (.hom and .hom.indiv) was plotted with R[94].

**Admixture characterisation and dating (MOSAIC)**

The MOSAIC v1.3.6 software[56] was run to characterise and date possible admixture events in the dataset. The phased haplotype input was obtained from the SHAPEIT2 .haps output files, using the R script `convert_from_haps.R`, available at https://csgitlab.ucd.ie/mst/mosaic/tree/master. The rest of the input files was prepared using custom bash scripts. The same genetic map was used as for phasing. The analyses were performed with R v3.6.1. Two Lozi individuals with recent non-African admixture were excluded from the analyses. Two- and three-way admixture models were tested for the five Zambian populations, with and without specifying source populations. The specified sources were Ju|'hoansi and Yoruba for two-way admixture; and Ju|'hoansi, Yoruba and Baka (Cameroon) for three-way admixture. We also tested four-way admixture with specified sources (Ju|'hoansi, Yoruba, Baka (Cameroon) and Amhara) for the two Zambian BaTwa populations. Additionally, two-way admixture was tested in the Baka from Cameroon (RHG), the Nzime (Bantu-speakers RHG neighbours from Cameroon), the southeastern Bantu-speakers (Bantu-speakers from South Africa), the !Xun (northern KS from Angola), the #Khomani and the Karretjie people (southern KS from South Africa). For the latter four populations, we also tested two-way admixture with specified sources (Ju|'hoansi and Yoruba). Confidence intervals for the estimated admixture dates were obtained by bootstrapping individuals. We compared the admixture time estimate obtained with MOSAIC for the southeastern Bantu-speakers to that obtained with a LD decay-based method (Supplementary Discussion).

The co-ancestry curves plots are in the GitHub repository[57]; see also Supplementary Figs. 16, 17, 20, 21, 22, 23.

**Uniparental markers (Y chromosome haplogroups)**

The Y chromosome variants were extracted with plink v1.90b4.9[83] from the original fileset for the BaTwa individuals. The dataset was filtered for variant and individual missingness (`–geno 0.1 –mind 0.1`), variants were renamed to the format `chr:position`, and variants with duplicate IDs were removed. The final dataset consisted of 2,487 variants in 33 males. The same procedure was applied to the Zambian agropastoralist populations, resulting in 2326 variants in 41 males.

Prior to haplogroup calling with SNAPPY v0.1[59], the variants in the dataset were flipped in order to match the strand in the reference genome. For this purpose, we created a plink fileset based on the reference file `pos_to_allele.txt` available with the software SNAPPY. In total, 918 sites overlapped between the dataset and the reference file, of which 44 had to be flipped. The haplogroups were called with SNAPPY and Python v2.7.6. The corresponding haplogroups in the ISOGG Y-DNA haplogroup tree (Y-DNA Haplogroup Tree 2019 V15.58 2020) were identified via their defining markers. We note that in two cases, the markers defining haplogroups in SNAPPY defined two haplogroups in the ISOGG tree; we refer to the haplogroups by the most precise name (see Supplementary Table 14 for the correspondence between different names for the same haplogroup).

We could not reliably call the mitochondrial DNA haplogroups: the variants on the H3Africa v1 callset were not informative enough to distinguish the main African mitochondrial DNA haplogroups[63].

## X chromosome to autosomes ancestry ratio

**Prepare the autosomes to mimic the X chromosome.** To limit biases, the autosomal data was processed to mimic the X chromosome in terms of genetic length and density of SNPs, as in ref. 66. The same genetic map was used as for phasing. First, we selected the autosomes that had a genetic length of $180 \pm 5$ cm or more (i.e., as long or longer than the X chromosome) and, if needed, cut the chromosomes to 180 cm (by finding the location closest to 180 cm and using plink `-to-bp` flag). Chromosomes 1–7 were cut while chromosomes 8, 9, 10 and 12 were kept unmodified. Chromosomes 11 and 13–22 were not used. The second step was to generate 100 sets of variants with the same SNP density like on the X chromosome (for which we had 3729 variants). For this we used a bash loop that (i) selected an autosome from the subset with the `shuf` command, and (ii) randomly selected 3729 positions on that chromosome. The resulting list was extracted with plink `-extract` flag.

**Run ADMIXTURE.** We ran supervised ADMIXTURE v1.3.0[54] on the 100 autosomal sets and on the X chromosomal set. The number of genetic cluster (K) was set to two and the source populations were the Ju|'hoansi (Khoe-San) and the Yoruba (western Africans). Twenty-five repeats were run for each input.

Prior to running ADMIXTURE for the X chromosome, we determined the sex of the individuals for which we did not already have the information. We ran plink `-check-sex` without parameters and plotted the resulting X chromosome inbreeding coefficients (F estimates). After visual inspection, we set thresholds and created a new fileset with information about the sex of all individuals with `-impute-sex 0.5 0.9`. Male X chromosome heterozygous calls were set to missing. We added the flag `-haploid="male:23"` to the ADMIXTURE command to specify that males are haploid on the X chromosome.

We used pong v1.4.7[95] to check that the runs had a single mode. We noted that this was not always the case when including the non-African populations which is the reason why we excluded the CEU and the Han from these analyses.

For each autosomal set and each iteration (i.e., a total of 2500 ADMIXTURE runs), we extracted the genetic membership proportion to each of the two sources, for each individual. We then averaged the proportion for each population across all runs and individuals in the population. For the 25 X chromosomal iterations, we did the same except that females' genetic memberships contributed twice as much as males'.

**Calculate female and male contributions and the ratios.** We calculated the female and male contributions using equations 22 and 23 in ref. 64, as well as the X chromosomal to autosomal ancestry ratio (by dividing the X chromosomal average by the autosomal average) (Supplementary Tables 11, 12).

### Reporting summary
Further information on research design is available in the Nature Portfolio Reporting Summary linked to this article.

## Data availability
The genotype data generated in this study for the Zambian BaTwa individuals have been deposited in the European Genome-Phenome Archive (EGA) under accession code EGA50000000364. The genotype data are available under restricted access for non-commercial projects addressing population and ancestry research questions. Access is granted only to the applicant and for a specific project.

Access can be obtained by applying to the EGA Data Access Committee EGAC50000000258. The genotype data for the Bemba, Lozi and Tonga[53] used in this study are available in the EGA database under accession code EGAS50000000006. The genotype data for the Dinka, Hadza and Sabue[13] used in this study are available in the NIH dbGAP repository under accession code phs001780.v1.p1 [https://www.ncbi.nlm.nih.gov/projects/gap/cgi-bin/study.cgi?study_id=phs001780.v1.p1]. The authorized NIH Data Access Committee (DAC) granted data access to CMS (date of approval: 2019-05-17). The genotype data for the Ba.Kiga, Baka (from Cameroon and Gabon), BaTwa (from Uganda), Bemba, Bongo, Nzebi and Nzime[87] used in this study are available in the EGA database under accession code EGAS00001000605. Data access was granted by the EGA DAC EGAC00001000139 (date of approval: 2021-05-07). The genotype data for the Amhara, Igbo, Mandinka, Oromo, Somali, Sotho and Zulu[89] used in this study are available in the EGA database under accession code EGAS00001000959. The genotype data for the Sara and Toubou[90] used in this study are available in the EGA database under accession code EGAD00010001103 under the standard Sanger publication policy. The ancient DNA data[19] used in this study is available upon request. The map used in Fig. 1a is available in the figshare database [https://doi.org/10.6084/m9.figshare.6396959.v1] (ref. 99 version 1). The map used in Fig. 1b is in the public domain (https://www.naturalearthdata.com).

## Code availability
Code for the analyses presented here can be accessed in the following GitHub repository: https://github.com/Gwennid/zambia_batwa[57].

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

## Acknowledgements

We are grateful to all individuals who participated in the study and for the permission to undertake this research in Zambia granted by the University of Zambia Biomedical Research Ethics Committee and the National Health Research Authority. We thank the staff from the Livingstone Museum who undertook the fieldwork and facilitated the translation of the participant information, informed consent forms, as well as a poster and summary of the study outcomes. We also thank the staff from the National Museums Board and the Department of National Parks and Wildlife who contributed to the fieldwork when the outcomes from the study were presented and discussed with the study participants. Genotyping was performed by the SNP&SEQ Technology Platform in Uppsala. The facility is part of the National Genomics Infrastructure supported by the Swedish Research Council for Infrastructures and Science for Life Laboratory, Sweden. The SNP&SEQ Technology Platform is also supported by the Knut and Alice Wallenberg Foundation. The computations were enabled by resources in projects SNIC 2020/2-10, SNIC 2018/8-397 and SNIC 2019/8-157 provided by the Swedish National Infrastructure for Computing (SNIC) at UPPMAX, partially funded by the Swedish Research Council through grant agreement no. 2018-05973. We thank Thijessen Naidoo for reviewing a draft of the

Y haplogroup section. This work was funded by the European Research Council (ERC) under the European Union's Horizon 2020 research and innovation programme (759933, C.M.S.), the Knut and Alice Wallenberg foundation (C.M.S., M.J.), and the Vetenskapsrådet/Swedish Research Council (2022-04642, 2018-05537, M.J.).

## Author contributions

L.B., M.J., G.M., C.S., and H.S. designed the research. G.M. performed the sampling. G.B. processed the data and performed the analyses. L.B. and G.B. wrote the manuscript with input from all authors. L.B., G.B., and G.M. presented and discussed the results with the populations. All authors read and approved the final manuscript.

## Funding

## Competing interests

The authors declare no competing interests.
