## [Peer Review File · Nature Communications]

BaTwa populations from Zambia retain ancestry of past hunter-gatherer groupsReviewers' Comments:

Reviewer #1:

Remarks to the Author:

The manuscript NCOMMS-23-11554, entitled "'BaTwa" populations from Zambia retain ancestry of past hunter-gatherer groups" by Gwenna Breton and colleagues, reports genome-wide SNP array data for five present-day human groups living in Zambia, studied at the genetic level for the first time.

These five groups include two BaTwa communities, composed of marginalized hunters and fishers, and three communities of agropastoralists. The term BaTwa (or BaCwa) is used by Bantu speakers to designate "the others", and often refer to distinctive groups of (former) hunter-gatherers (HGs) living in different African countries, such as Uganda, Rwanda, Burundi and DRC. Very few studies have assessed whether the Bantu-speaking BaTwa are descended from either Bantu-speaking migrants who acquired distinctive cultural practices, or from autochthonous hunter-gatherers who admixed with Bantu-speaking newcomers (Patin et al., *Nat Commun* 2014; Perry et al., *PNAS* 2014). This question is particularly relevant given our very limited knowledge of the genetic structure of African populations before the Bantu expansions, which considerably altered the genetic landscape of sub-Saharan Africa, and the difficulties to obtain ancient DNA in these regions.

The authors show that the BaTwa from Zambia, as well as neighboring Bantu speaking agropastoralists, have retained ancestry from a population relating to present-day hunter-gatherers from central (RHG) and southern (KS) Africa. They estimate that admixture between this ancestral source and Bantu-speaking migrants occurred 15-45 generations ago in Zambia, i.e., few generations after the earliest archaeological evidence for the arrival of farmers in these regions. The authors also report evidence for sex-biased admixture, indicating a higher proportion of western central African males than females among the ancestors of Zambians. This new study extends previous findings based on ancient DNA, indicating that sub-Saharan Africa was peopled by groups of hunter-gatherers who formed a three-way cline of ancestry found in present-day hunter-gatherers from central (RHG), eastern (eHG) and southern Africa (KS) (Skoglund et al., *Cell* 2017; Lipson et al., *Nature* 2022).

The manuscript is very well written, analyses are sound and carefully interpreted. I have three major comments that, in my view, should improve the study.

1. I will start with a very general comment about the authors' interpretation of ancestry-based analyses. The authors focus strongly their conclusions on the BaTwa, by stating, for example, that "the BaTwa are descendants of HG populations" (l. 469); however, these conclusions also hold for Zambian agropastoralists. Although it is clear from the authors' analyses that all Zambian populations investigated here are descended – in part – from local HGs, the authors only highlight the BaTwa as the descendants of local HGs, implying that agropastoralists are not. This can be problematic in several regards. First, this implies that most of the ancestors of the BaTwa were local HGs, which is not the case. Second, the authors' interpretation relies implicitly on the assumption that the BaTwa have always been marginalized groups who practice hunting, but this may not be the case. We can easily imagine that the BaTwa from Zambia were Bantu-speaking farmers who admixed with local HGs and started practicing hunting (see e.g., the Mikea or the Ogiek, Pierron et al., *PNAS* 2014; Scheinfeldt et al., *PNAS* 2019). Third, the fact that Zambian agropastoralists also carry local HG ancestry is actually of great interest, as it reinforces the idea that local HGs were not small isolated groups and it further highlights how the genetic history of present-day Zambia differs from that of Malawi. Fourth, there may be some uncertainty regarding the ethnic origin of the sampled individuals, since Zambians are reluctant to self-identify as BaTwa. Fifth, the authors' conclusions may reinforce the idea that the BaTwa are genetically distinct from other Zambian populations, which might contribute to their stigmatization. For these different reasons, I suggest that the authors report more systematically the results for agropastoralists, discuss further these results, and avoid statements that imply that the BaTwa are the only descendants of local HG populations.

2. The authors report evidence that Zambians descend from a three-way admixture, a model where single f -statistics can be difficult to interpret (e.g., correlation of f_4 ratios in Figure 3). Interpretability

can be improved by modelling all f-statistics jointly, by searching for the best-fit admixture graphs. I suggest that the authors do so, with the `find_graphs` function of ADMIXTOOLS2.

3. In their analyses, the authors represent RHGs by the Baka, who are RHGs from western Central Africa. However, other RHG groups live today in the eastern part of the African Equatorial Rainforest, including the BaTwa from Uganda and the "Mbuti" (Asua or Efe) from DRC (Patin et al., Nat Commun 2014). These groups may represent a better source of RHG ancestry than the Baka in Zambians, as suggested by ADMIXTURE results at $K > 8$ (Figure S5). The use of the Mbuti (who are not admixed with Bantu-speaking farmers and have been whole genome sequenced) instead of the Baka may substantially affect the authors' results, including admixture proportion estimates, f3 admixture tests, Figure 5 and the closest proxies for the minor ancestry source in the BaTwa from Zambia. I invite the authors to merge their data with HGDP gVCFs (<ftp://ngs.sanger.ac.uk/production/hgdp>), remove inconsistent alleles between datasets and explore the affinities of Zambians to the Mbuti.

Minor comments:

- In their analyses, the authors represent eHG ancestry by the Hadza (or by ancient HGs from Malawi). However, the Hadza are admixed with western central Africans. I am curious to see if f-statistics change when using the Sabue or Mota (Bayira) instead.
- L.56: References to linguistic studies would be relevant here (Grollemund et al., PNAS 2015; Kolie et al., PNAS 2022).
- L.71: The authors could cite here Padilla-Iglesias et al., PNAS 2022.
- L.121: Please replace resembles by resemble.
- L.173: Why projecting the Zambians?
- L. 184-186: Where are these results shown?
- L.212-214: I was expecting a statistical test here.
- L.238: Please replace Usning by Using.
- L. 353: Reporting some numbers in the main text would be useful.
- L.367: Please replace analyzes by analyses.
- L.404: Please replace whos by whose.
- L.434-437: This scenario does not seem to be supported by admixture dates and archaeological data. Can the authors comment?
- L.460: Please replace agropastoralist by agropastoralists.
- L.546-547: Can the authors rephrase to clarify that the overlapping variants refer to variants in HWD, and not all variants?
- L.672: Pseudo-haploid?
- L.701: Please replace follow by follows.
- L.759: Please rephrase "We used the same recombination map like for phasing."

Reviewer #2:

Remarks to the Author:

In the paper "BaTwa populations from Zambia retain ancestry of past hunter-gatherer groups", Breton and colleagues analyze genomic data from five populations from Zambia to reconstruct admixture dynamics between Bantu type genetic substrate and non-Bantu genetic substrate. The non-Bantu genetic substrate is linked to a hunter-gatherer substrate, and is found at highest proportions in the BaTwa populations, which carry a history of discrimination from Bantu groups, are characterized by hunting and fishing subsistence, and are associated with an autochthonous regional origin. The paper provides new genome-wide data from a relevant region of sub-Saharan Africa, which is traditionally not well represented in genetic studies (at least in comparison to Eurasia). The analysis of the possible HG substrates is relevant.

The focus is quite local, for the attention on the two target populations. Their relevance could be better contextualized, or the results could be further discussed in light of a broader, sub-Saharan

picture, to get the attention of a broader readership. This is mentioned for example in line 46, but does not emerge enough through the paper.

The paper bases most of its conclusions on ADMIXTURE analysis, and on the comparison of frequencies of the ancestries of interest. This requires further contextualization, as appears as the main result of the paper taking space in the discussion. Some points of weakness can be improved.

The general admixture scenario should be discussed for all the Zambian Bantu groups, not only forcing the focus on the Batwa as their history is completely different from the other neighboring groups. In the paper, they seem to be characterized only by higher frequencies of admixture. I see the difference is tested, but still does not justify a completely different demographic story.

No Cross-Validation error is reported for each K examined in the ADMIXTURE analysis.

I suggest that the paper could benefit from more independent analysis to assess the differences in admixture between the Zambia populations and the possible sources. There are no direct tests to model the minor genetic component as an admixture of Khoisan and Rain forest HG ancestries. The analysis of the Rain forest HG component in Batwa seems a bit stretched. Line 430: why the Rain forest HG component should be the sole source of HG ancestry in the Batwa?

The fact that only the Kafue admixture is confirmed with f3 analysis is not further commented on.

The possible admixture combinations are not further tested with f4. F4 ratio tests are performed, but this section could benefit from further clarification on what was tested and how the significance is measured.

Admixture dates from MOSAIC are commented on only for the two Batwa populations of interest. What is the general picture for Zambia? And outside Zambia? Is MOSAIC sensitive to multiple pulses of admixture? Would MOSAIC register only the most recent pulse of admixture?

How can we confidently link the minor admixture component to a HG population? The safest way to describe it could be a non-Bantu genetic substrate. Is the HG component anchored with archaeological data? Which type of archaeological evidence is used to anchor the passage from HG to agriculturalist? What about the pastoralist wave from East Africa before the Bantu spread, how does it affect these regions in Zambia?

About the Batwa groups. What is the definition of Batwa? 1) "Ba" is the plural prefix when referring to people, "Twa", in Bantu languages, or 2) "people who always move" or "the others"? is it an exonym? What about the Batwa outside of Zambia? In the methods section, it appears as the Batwa do not like to be recognized with this name. Is it ok still to use this denomination for the paper? Is it derogatory? If the results of this paper would be discussed with Batwa participants, how would they like to be represented? I understand that discussing the results with the communities is not always possible because of time and resource constraints, but I wonder if the paper reports their history and their identity appropriately.

The paper should mention why no mtDNA data is reported in the results, not only in the methods. It would be an obvious question for the reader, why there is no mention of mtDNA? It is peculiar that in the chip there is data for Y chromosome but not for mtDNA.

Line 390: elaborate on the language scenario, why would it match the admixture profiles?

In the results section, explicitly mention the type of analysis with the software when you introduce them: ADMIXTURE, and MOSAIC, for example.

(Author response in blue)

REVIEWER COMMENTS

Reviewer #1 (Remarks to the Author):

The manuscript NCOMMS-23-11554, entitled ““BaTwa” populations from Zambia retain ancestry of past hunter-gatherer groups” by Gwenna Breton and colleagues, reports genome-wide SNP array data for five present-day human groups living in Zambia, studied at the genetic level for the first time. These five groups include two BaTwa communities, composed of marginalized hunters and fishers, and three communities of agropastoralists. The term BaTwa (or BaCwa) is used by Bantu speakers to designate “the others”, and often refer to distinctive groups of (former) hunter-gatherers (HGs) living in different African countries, such as Uganda, Rwanda, Burundi and DRC. Very few studies have assessed whether the Bantu-speaking BaTwa are descended from either Bantu-speaking migrants who acquired distinctive cultural practices, or from autochthonous hunter-gatherers who admixed with Bantu-speaking newcomers (Patin et al., Nat Commun 2014; Perry et al., PNAS 2014). This question is particularly relevant given our very limited knowledge of the genetic structure of African populations before the Bantu expansions, which considerably altered the genetic landscape of sub-Saharan Africa, and the difficulties to obtain ancient DNA in these regions.

The authors show that the BaTwa from Zambia, as well as neighboring Bantu speaking agropastoralists, have retained ancestry from a population relating to present-day hunter-gatherers from central (RHG) and southern (KS) Africa. They estimate that admixture between this ancestral source and Bantu-speaking migrants occurred 15-45 generations ago in Zambia, i.e., few generations after the earliest archaeological evidence for the arrival of farmers in these regions. The authors also report evidence for sex-biased admixture, indicating a higher proportion of western central African males than females among the ancestors of Zambians. This new study extends previous findings based on ancient DNA, indicating that sub-Saharan Africa was peopled by groups of hunter-gatherers who formed a three-way cline of ancestry found in present-day hunter-gatherers from central (RHG), eastern (eHG) and southern Africa (KS) (Skoglund et al., Cell 2017; Lipson et al., Nature 2022).

The manuscript is very well written, analyses are sound and carefully interpreted. I have three major comments that, in my view, should improve the study.

1. I will start with a very general comment about the authors' interpretation of ancestry-based analyses. The authors focus strongly their conclusions on the BaTwa, by stating, for example, that “the BaTwa are descendants of HG populations” (l. 469); however, these conclusions also hold for Zambian agropastoralists. Although it is clear from the authors' analyses that all Zambian populations investigated here are descended – in part – from local HGs, the authors only highlight the BaTwa as the descendants of local HGs, implying that agropastoralists are not. This can be problematic in several regards. First, this implies that most of the ancestors of the BaTwa were local HGs, which is not the case. Second, the authors' interpretation relies

implicitly on the assumption that the BaTwa have always been marginalized groups who practice hunting, but this may not be the case. We can easily imagine that the BaTwa from Zambia were Bantu-speaking farmers who admixed with local HGs and started practicing hunting (see e.g., the Mikea or the Ogiek, Pierron et al., PNAS 2014; Scheinfeldt et al., PNAS 2019). Third, the fact that Zambian agropastoralists also carry local HG ancestry is actually of great interest, as it reinforces the idea that local HGs were not small isolated groups and it further highlights how the genetic history of present-day Zambia differs from that of Malawi. Fourth, there may be some uncertainty regarding the ethnic origin of the sampled individuals, since Zambians are reluctant to self-identify as BaTwa. Fifth, the authors' conclusions may reinforce the idea that the BaTwa are genetically distinct from other Zambian populations, which might contribute to their stigmatization. For these different reasons, I suggest that the authors report more systematically the results for agropastoralists, discuss further these results, and avoid statements that imply that the BaTwa are the only descendants of local HG populations.

Authors: Thank you for this comment and observation. We completely agree that it would be wrong to interpret our results/text as stating that the BaTwa are (completely) descendants of autochthonous groups. Although we have been quite clear on that it is a minor fraction of the ancestry that is autochthonous (e.g. abstract: "*We found that both BaTwa populations harbor a hunter-gatherer-like genetic component, representing ~19% (Bangweulu) and ~31% (Kafue), of their genetic ancestry, while the rest of their ancestry...*"), there are other instances that could be misinterpreted as pointed out. We have edited the text throughout to make it clear that the BaTwa have mixed ancestry, where the minor fraction of ancestry comes from autochthonous groups (e.g. lines 38-39, 167-168, 222, 470-474, 479-483, 492-495).

Thank you also for pointing out the impression of a biased focus on the BaTwa over the agropastoralists. We have added (and moved from the supplement) a richer discussion on the agropastoralist groups and the interpretation of their (also) mixed ancestry (e.g. line 487 and onward). Having said that, we do still believe that the BaTwa deserves a greater attention as they are severely underrepresented in studies of human diversity in Africa. There are for instance several studies with large numbers of Bantu-speaking groups of West African ancestry that describes the history of the groups that were linked to the 'Bantu-expansion' (e.g. Patin et al Science 2017, Fortes-Lima et al., Biorxiv 2023).

Reluctance to self-identify was only experienced with the BaTwa on the islands of Lake Bangweulu. They were reassured when historical explanations of their potential early ancestry were explained and that they were as important as any other group in Zambia. This recognition of importance and equality resulted in a willingness to discuss ancestry and identify those in the community with the BaTwa parents and grandparents. The Kafue BaTwa were more assured and assertive of their ancestry. The process of recruiting participants is described in Methods, under the subheading "Sampling and genome-wide SNP typing".

2. The authors report evidence that Zambians descend from a three-way admixture, a model where single f-statistics can be difficult to interpret (e.g., correlation of f4 ratios in Figure 3). Interpretability can be improved by modelling all f-statistics jointly, by searching for the best-fit

admixture graphs. I suggest that the authors do so, with the `find_graphs` function of ADMIXTOOLS2.

Authors' response: We realize that the text was not always clear on this topic. We revised the wording to avoid suggesting a three-way admixture. Rather, we think that our data supports a two-way admixture, with one source being intermediate between two hunter-gatherer genetic backgrounds (especially Khoe-San and RHG). The fact that the HG component appears as "admixed" is not necessarily a consequence of admixture, more likely a consequence of only having access to distantly related KS and RHGs. At lines 40, 236-239, 299-302, 417, 514-515, 526-527 we rephrased "mix" or "mixture" as "intermediate" when referring to the HG component, or specifically say 2-way admixture when referring to the west African and the HG admixture.

We also followed the suggestion of using `find_graphs` to systematically search through multiple f-test-values. Overall, we find that a 2-way admixture between a west African source and a HG source is supported for Kafue, but for the other 4 Zambian groups, we find a single west African source (or for Lozi a 3% HG contribution in a 2-way admixture). This observation is in line with the f3-test being significant only for the Kafue for a 2-way admixture, likely due to a combination of (i) there is no good proxy population for the HG ancestry component in the Zambian populations (ii) the HG component is modest-small, (iii) limited power of f3-tests. A summary of the `find_graphs` analysis can be found in the Appendix at the end of this document. We have included the result of the `find_graphs` search in lines 250-258, and as supplementary figure S19.

3. In their analyses, the authors represent RHGs by the Baka, who are RHGs from western Central Africa. However, other RHG groups live today in the eastern part of the African Equatorial Rainforest, including the BaTwa from Uganda and the "Mbuti" (Asua or Efe) from DRC (Patin et al., Nat Commun 2014). These groups may represent a better source of RHG ancestry than the Baka in Zambians, as suggested by ADMIXTURE results at $K > 8$ (Figure S5). The use of the Mbuti (who are not admixed with Bantu-speaking farmers and have been whole genome sequenced) instead of the Baka may substantially affect the authors' results, including admixture proportion estimates, f3 admixture tests, Figure 5 and the closest proxies for the minor ancestry source in the BaTwa from Zambia. I invite the authors to merge their data with HGDP gVCFs (<ftp://ngs.sanger.ac.uk/production/hgdp>), remove inconsistent alleles between datasets and explore the affinities of Zambians to the Mbuti.

Authors' response: Thank you for your suggestion of representing RHG by an eastern RHG population. We used the BaTwa from Uganda as a proxy for eastern rainforest hunter-gatherers and we repeated the following analyses: projected PCA (as in Figure 1c); f4 ratio test (as in Figure 3); f3 test. We found that the results are very similar irrespective of the choice of source RHG population (the eastern RHG population BaTwa from Uganda or the Baka). These results are now reported in the manuscript (projected PCA: Figure S15, f4 ratio: Figure S16, f3 test results added to Table S4). The results for the f4 ratio with the Baka are significantly correlated to the results with the Ugandan BaTwa (Spearman rho correlation p-value $< 2.2e-16$, rho = 1).

Since the MOSAIC analysis is unsupervised, the Ugandan BaTwa is already included in the pool of possible proxies for the minor ancestry source (Table S10), but the better proxy was found to be Baka.

Minor comments:

- In their analyses, the authors represent eHG ancestry by the Hadza (or by ancient HGs from Malawi). However, the Hadza are admixed with western central Africans. I am curious to see if f-statistics change when using the Sabue or Mota (Bayira) instead.

Authors' response: Thank you for the suggestion. We repeated the f3 test and the f4 ratio test with the Sabue representing the eHG instead of the Hadza. This gave very similar results. The results are presented in Figure S16 and Table S4. The f4 ratio test results were significantly correlated whether using the Hadza or the Sabue (Spearman rho correlation p-value < 2.2e-16, rho = 1).

- L.56: References to linguistic studies would be relevant here (Grollemund et al., PNAS 2015; Kolie et al., PNAS 2022).

Authors' response: thank you for the suggestion. We added the references.

- L.71: The authors could cite here Padilla-Iglesias et al., PNAS 2022.

Authors' response: thank you for the suggestion; the reference was added.

- L.121: Please replace resembles by resemble.

Authors' response: fixed

- L.173: Why projecting the Zambians?

Author response: we showed a PCA where the Zambian populations projected as it recapitulates, in two dimensions, patterns seen in more dimensions when the Zambian populations are not projected. The results without projection, for the same individuals like in Figure 1C, are shown in Figure S15.

- L. 184-186: Where are these results shown?

Authors' response: the results of the projected PCA where the Nzime, a RHG neighbor population who speak a Bantu language rather than using the Yoruba, who speak Niger Congo (non-Bantu) language, are shown in Figure S16.

- L.212-214: I was expecting a statistical test here.

Authors' response: we added the standard error of the mean for the agropastoralist populations as well, lines 224-225 (the standard errors are non overlapping, except between the Lozi and the Bemba). See values for other populations in Supplementary Table 3.

- L.238: Please replace Usning by Using.

Authors' response: fixed

- L. 353: Reporting some numbers in the main text would be useful.

Authors' response: the text was edited to include numbers (lines 377-382).

- L.367: Please replace analyzes by analyses.

Authors' response: fixed

- L.404: Please replace whos by whose.

Authors' response: fixed

- L.434-437: This scenario does not seem to be supported by admixture dates and archaeological data. Can the authors comment?

Authors: The admixture dates and archaeological dates correspond closely in the immediate vicinity of each sampling locality. The admixture dates fall within Southern Hemisphere calibrated radiocarbon ages for the earliest known Early Iron Age sites in the Kafue plains ~1525 AD (98.9% probability) at Sebanzi Hill. The Kafue admixture date of 450 BP equates with a radiocarbon calendar age of 1500 AD. For Lake Bangweulu the calibrated radiocarbon age of ~682 cal AD (94.5% probability) at Samfya is slightly older than the estimated admixture date but confirms an earlier contact in this part of Zambia (L.405-418.)

- L.460: Please replace agropastoralist by agropastoralists.

Authors' response: fixed

- L.546-547: Can the authors rephrase to clarify that the overlapping variants refer to variants in HWD, and not all variants?

Authors' response: rephrased

- L.672: Pseudo-haploid?

Authors' response: right, thank you!

- L.701: Please replace follow by follows.

Authors' response: fixed

- L.759: Please rephrase "We used the same recombination map like for phasing."

Authors' response: rephrased

Reviewer #2 (Remarks to the Author):

In the paper "BaTwa populations from Zambia retain ancestry of past hunter-gatherer groups", Breton and colleagues analyze genomic data from five populations from Zambia to reconstruct admixture dynamics between Bantu type genetic substrate and non-Bantu genetic substrate. The non-Bantu genetic substrate is linked to a hunter-gatherer substrate, and is found at highest

proportions in the BaTwa populations, which carry a history of discrimination from Bantu groups, are characterized by hunting and fishing subsistence, and are associated with an autochthonous regional origin. The paper provides new genome-wide data from a relevant region of sub-Saharan Africa, which is traditionally not well represented in genetic studies (at least in comparison to Eurasia). The analysis of the possible HG substrates is relevant.

1. The focus is quite local, for the attention on the two target populations. Their relevance could be better contextualized, or the results could be further discussed in light of a broader, sub-Saharan picture, to get the attention of a broader readership. This is mentioned for example in line 46, but does not emerge enough through the paper.

Authors' response: Similar to reviewer 1's suggestion of a broader focus on all 5 Zambian populations, we have included a more extensive discussion of the agropastoralists (lines 481-503). In the discussion (and in the introduction), we have further expanded on the scope to highlight the methodological context of our study (that studying isolated groups is one of 2 possible approaches for genetic research to help decipher the human past of Africa, lines 391-407). See also our response to reviewer 1's comment 1.

2. The paper bases most of its conclusions on ADMIXTURE analysis, and on the comparison of frequencies of the ancestries of interest. This requires further contextualization, as appears as the main result of the paper taking space in the discussion. Some points of weakness can be improved.

Authors' response: There are several different analyses conducted for this paper, including model-free approaches (PCA, RoH), model-based approaches (f3, f4-ratio, Fst, admixture graph searchers – MOSAIC and find_graphs). We have expanded the Results sub-section "Characterizing the nature of the admixture in Zambian populations" which is based on other analyses than ADMIXTURE, and it is about 100 lines long (compared to approx 40 lines describing the ADMIXTURE results).

We would be very willing to improve upon "some points of weakness", but we would need some more detailed information in order to know what those "points of weakness" would be. Perhaps our revised version has dealt with these items.

3. The general admixture scenario should be discussed for all the Zambian Bantu groups, not only forcing the focus on the Batwa as their history is completely different from the other neighboring groups. In the paper, they seem to be characterized only by higher frequencies of admixture. I see the difference is tested, but still does not justify a completely different demographic story.

Authors' response: thank you for pointing this out. See the answer to reviewer 1, point 1. In particular, we expanded the results section to mention the agropastoralists where we did not do so earlier, and we expanded the discussion about the agropastoralists.

4.No Cross-Validation error is reported for each K examined in the ADMIXTURE analysis.

Authors' response: We do not believe that any single choice of K is more relevant than others, rather, the hierarchical combination of going from K=2 to greater values and examining consistency across multiple replicates and checking the likelihood reveals the most information from the data and analyses. However, we have added the Cross-Validation error in the Supplement for reference (Figure S19).

5.I suggest that the paper could benefit from more independent analysis to assess the differences in admixture between the Zambia populations and the possible sources. There are no direct tests to model the minor genetic component as an admixture of Khoisan and Rain forest HG ancestries. The analysis of the Rain forest HG component in Batwa seems a bit stretched. Line 430: why the Rain forest HG component should be the sole source of HG ancestry in the Batwa?

Authors' response: Thank you for pointing out this observation, which is related to reviewer 1's comment 2. Please also see our response to that question.

Specifically, we tested a three-way admixture in MOSAIC with the Yoruba, Ju|'hoansi and Baka as ancestries (Supplementary Fig. 10 and 11), and discuss the results lines 1279-1295. The co-ancestry curves suggest that a single admixture event is a better fit; moreover, the two minor ancestries colocalize on the karyograms. We have now added an additional search across admixture graphs, which comes back supporting a 2-way admixture (see response to rev 1's comment 2). We further moved some of the results related to the question of whether the RHG component can be distinguished from the KS component and of its origin, including main figure 5 (now Supplementary Figure S14), to the supplements (lines 1296-1317), and edited the discussion accordingly (lines 464-471).

6.The fact that only the Kafue admixture is confirmed with f3 analysis is not further commented on.

Authors' response: we expanded the paragraph about the f3 test results (lines 247-251).

7.The possible admixture combinations are not further tested with f4. F4 ratio tests are performed, but this section could benefit from further clarification on what was tested and how the significance is measured.

Authors' response: We have reworked the section on "Characterizing the nature of the admixture in Zambian populations", and please see also response to reviewer 1, comment 2. We added the full results, with the value of alpha, the standard error and the Z-score, in Supplementary Tables S13 and S14. Standard errors for the f4 ratio tests are given in the text line 267 and shown in Figure 3, S6 and S18. We can interpret the results of the f4 ratio test where the Ju|'hoansi is the tested admixture source (Figure 3a) as follow: the populations with a value of 1-alpha significantly larger than 0 (i.e. from BaTwa (Kafue) to Igbo) share a common

ancestry with the branch represented by the Ju|'hoansi. Similar interpretations can be made of the other results.

8. Admixture dates from MOSAIC are commented on only for the two Batwa populations of interest. What is the general picture for Zambia? And outside Zambia? Is MOSAIC sensitive to multiple pulses of admixture? Would MOSAIC register only the most recent pulse of admixture?

Authors' response: We have expanded the discussion on the agropastoralists (lines 494-516; part of this was included as Supplementary Discussion in the first version of the manuscript) and contrast to southeastern Bantu-speakers from South Africa (lines 1319-1326).

MOSAIC can characterize admixture with more than two admixing populations, and is not sensitive to the most recent event only (Salter-Townshend and Myers, 2019). In the Supplementary Notes (lines 1279-1295), and Supplementary Fig. 10-13, we present and discuss the results for three and four-way admixture/multiple pulses with MOSAIC (for the Zambian BaTwa).

9. How can we confidently link the minor admixture component to a HG population? The safest way to describe it could be a non-Bantu genetic substrate. Is the HG component anchored with archaeological data? Which type of archaeological evidence is used to anchor the passage from HG to agriculturalist? What about the pastoralist wave from East Africa before the Bantu spread, how does it affect these regions in Zambia?

Author: The archaeological evidence for HG in Zambia takes the form of a widespread microlithic Later Stone Age tool industry which is widespread across the region from 20,000 years ago. This industry continues long after the arrival of agriculturalists in some parts of Zambia, notably to the east in the Luangwa Valley (Colton 2009), in northern Zambia (Musonda 1987, Fletcher et al. 2022) with evidence of interaction between HG and agriculturalists in the form of the introduction of ceramics into the material culture of HGs. Dietary isotope data extracted from human skeletal material at Mumbwa Caves, 90 km north of the Kafue plains, shows the co-existence of a distinctive forager pattern of food consumption alongside those eating domesticated plants until the late 19th c (Steyn et al. 2022). A third indicator of differences in material culture is the prevalence of red and black geometric rock art associated with Later Stone Age sites in Zambia and Malawi; this tradition is not reflected in the white animal and human imagery associated with historic and recent farming communities in the region (Smith 1996). Where the two traditions co-occur, the white clay pictographs overlie the red geometrics suggesting a temporal and cultural change.

We further examined the genotypes of the Zambian populations at two mutations associated with lactase persistence and frequent in East Africa (Segurel and Bon, Annual Review of Genomics and Human Genetics 2017), -14.010:G>C (rs145946881) and -13.907:C>G (rs41525747). -14.010:C is also found in relatively high frequencies in some southern African populations, such as the Nama. We found one copy of -14.010:C in the BaTwa from Bangweulu. All other Zambian individuals are homozygous for -14.010:G and -13.907:C – the

non-lactase persistent variants. Other lactase persistence associated variants were not included in the chip.

10. About the Batwa groups. What is the definition of Batwa? 1) "Ba" is the plural prefix when referring to people, "Twa", in Bantu languages, or 2) "people who always move" or "the others"? Is it an exonym? What about the Batwa outside of Zambia? In the methods section, it appears as if the Batwa do not like to be recognized with this name. Is it ok still to use this denomination for the paper? Is it derogatory? If the results of this paper would be discussed with Batwa participants, how would they like to be represented? I understand that discussing the results with the communities is not always possible because of time and resource constraints, but I wonder if the paper reports their history and their identity appropriately.

Authors: BaTwa is an exonym in Zambia (as it is in Rwanda and Burundi), and Ba is the plural of MuTwa which refers to individuals. The descriptions of the BaTwa lifestyle ('always on the move', 'the others') were recorded by Barham from conversations with Bemba speakers living near the Bangweulu BaTwa. Reluctance to self-identify was only experienced with the BaTwa on the islands of Lake Bangweulu. They were reassured when historical explanations of their potential early ancestry were explained and that they were as important as any other tribe in Zambia. This recognition of importance and equality resulted in a willingness to discuss ancestry and identify those in the community with the BaTwa parents and grandparents. The Kafue BaTwa were more assured and assertive of their ancestry.

11. The paper should mention why no mtDNA data is reported in the results, not only in the methods. It would be an obvious question for the reader, why there is no mention of mtDNA? It is peculiar that in the chip there is data for Y chromosome but not for mtDNA.

Authors' response: we added a mention of this in the results section, lines 367-369.

12. Line 390: elaborate on the language scenario, why would it match the admixture profiles?
Authors' response: we rephrased the sentence to make it clearer, lines 427-431.

13. In the results section, explicitly mention the type of analysis with the software when you introduce them: ADMIXTURE, and MOSAIC, for example.

Authors' response: we included more explicit mention of the softwares, lines 198, 264, 285.

# Appendix:

## Find_graphs results

**BaTwa from Kafue**
m=3 was significantly better than m=2
In the best graph, the BaTwa from Kafue get 81% from a node most closely related to the Yoruba (ancestral to the Yoruba), and 19% from a neighboring group to the Juhoansi.
See suppl figure S19.

**BaTwa from Bangweulu**
m=3 was not significantly better than m=2. m=2 was significantly better than m=1.
Here, the BaTwa from Bangweulu are not admixed. The Baka and the Hadza are. The BaTwa from Bangweulu are quite basal, and sister group to the Yoruba.

**Bemba**
m=3 was not significantly better than m=2. m=2 was significantly better than m=1.
The graph is quite similar to the graph for the BaTwa from Bangweulu. The Bemba are quite basal. The Baka and Hadza are admixed. The Bemba branch off before the Yoruba (short branch though).

Tonga

m=3 was not significantly better than m=2. m=2 was significantly better than m=1.
The graph is quite similar to the Bangweulu & Bemba graphs.

Lozi

m=3 was significantly better than m=2

The Lozi get 95% from a common ancestor with the YRI; the common ancestor gets 3% from close to the root. The 5% remaining come from a sister group to the Ju|'hoansi, which is also ancestral to the Hadza (with long branch though).

We ran find_graphs again, after removing the two individuals with recent non-African admixture. $m=3$ is significantly better than $m=2$. Here, the three admixed populations are the Lozi, the Ju|'hoansi and the Hadza. The Lozi get 97% from a direct ancestor of the Yoruba, and 3% from a source that contributes most of the Ju|'hoansi ancestry and some of the Hadza ancestry.

Reviewers' Comments:

Reviewer #1:

Remarks to the Author:

The authors have satisfactorily addressed my comments.

I have three minor comments and have noticed few remaining typos:

- I find it interesting that most Eastern (Ba.Kiga, Luhya) and Southern (Sotho, Zulu) Bantu speakers have apparently no (or at least lower) western RHG (Baka-like) ancestry than Zambians. If confirmed (f4 ratios...), this could further suggest that RHG-like ancestry in Zambians was local and not brought by the Bantu expansion.
- L. 378-383: I suggest that the authors report the male and female contribution from West Africans in Zambians, instead of that of KS HGs. This would avoid negative values; male contributions slightly above 1 may be rounded to 1.
- L. 775: I suggest the authors test 4 admixture events (in case RHG > Zambian BaTwa is supported).
- L. 125: Replace "resembles" by "resemble"
- L. 158: Remove the first "more"?
- L. 197: Replace "have" by "has"
- L. 312: Replace "that" by "who"
- L. 556: Replace "African" by "Africa"
- L.761: Replace "admixture" by "admixture"
- L. 770: Replace "follow" by "follows"
- L. 778-780: Replace "was" by "were" or add "model"
- L. 891: Data and code are not made available yet.

Reviewer #3:

Remarks to the Author:

In this revised manuscript, Breton et al. combined data genetic data from five Zambian populations (including Bangweulu and Kafue "BaTwa" populations) with data from 39 other African populations to infer demographic history and historical patterns of admixture. Note that I was not one of the original reviewers, and in reviewing this paper I have opted to avoid groupthink and take a fresh perspective (i.e., I did not focus on the response to reviewer comments). Overall, this is a solid paper that is well-suited for a journal like Nature Communications. The authors have done a thorough job in their population genetics analyses, and they have employed the appropriate computational approaches. The writing is easy to follow and most figures and tables are easy to interpret (see below for minor exceptions). On the whole, I liked this paper and feel that it makes a valuable contribution to the anthropological genetics literature. However, minor revisions are still required for it to be suitable for publication. These changes involve some changes in wording, double-checking the results shown in Figure 3, briefly addressing how SNP ascertainment may bias their results, and ensuring that data and code availability standards are upheld. Additional comments and suggestions are listed below.

Abstract (page 3, lines 40-41): The authors should be careful about using phrase "does not match any living group" when referring to the hunter-gatherer ancestry component found in BaTwa genomes, especially in light of the Taíno genomics controversy that occurred over a decade ago. In addition, it would be good to include details about sample sizes in the abstract, particularly the number of novel samples that have been genotyped as part of this study.

Introduction: Multiple papers in the past three years have referred to the Sabue people of Ethiopia as the Chabu, and it might be good to mention this alternative label in this manuscript.

Figure 3 and lines 281-285: It is quite surprising that the patterns are so similar in panels A and B in this figure. Similarly, the values of Spearman's rho that are mentioned in lines 281-285 are striking enough to provoke a little skepticism (1.0 and -1.0). Why are these trends so similar for each of the populations described in Figures 3 and S6? The authors are encouraged to add a couple more sentences to this manuscript to help explain these striking patterns. Also, although alpha statistics from f4 ratio tests are appropriate, readers may be confused by negative values of "estimated fractions of J Jul'hoansi/Baka/Hadza ancestry" in Figures 3 and S6.

Figure S6: The Hadza are from Tanzania, not Ethiopia, yes?

Figure 4: What was the minimum RoH length used to generate this figure? 1Mb? In any case, these detail should be included in the figure legend.

The supplemental figures that depict co-ancestry curves (Figures S8, S9, S10, S11, S12, and S13) are missing axis labels. It also might be good to expand the figure legends of these supplemental figures to assist the reader. As presently shown, it is not immediately clear how one should interpret these curves.

Page 12, line 244: How much affinity? It would be good to quantify this.

Page 13, line 278: Should this sentence refer to admixture BETWEEN or INTO populations?

Page 15, lines 307-315: Given that this manuscript uses data from the Illumina H3Africa array, Fst statistics are likely to be affected by ascertainment bias (particularly for populations like the Bangweulu and Kafue). The authors are encouraged to mention this point and include any relevant citations.

Page 17, lines 378-380: It would be good to either report the actual amounts of X chromosome and autosome ancestry or include confidence intervals for female contributions (especially in light of the 2023 AJHG paper by Pfennig et al. about sex-biased admixture).

Please ensure that the EGA accession numbers are included in the final/next version of this manuscript. Also, the GitHub repository link listed on page 39 did not work.

Author response in blue

REVIEWERS COMMENTS

Reviewer #1 (Remarks to the Author):

The authors have satisfactorily addressed my comments.

I have three minor comments and have noticed few remaining typos:

- I find it interesting that most Eastern (Ba.Kiga, Luhya) and Southern (Sotho, Zulu) Bantu speakers have apparently no (or at least lower) western RHG (Baka-like) ancestry than Zambians. If confirmed (f4 ratios...), this could further suggest that RHG-like ancestry in Zambians was local and not brought by the Bantu expansion.

Authors: Figure 3b shows the estimated admixture proportion from a western RHG (Baka) population into, among other, the eastern and southern Bantu-speakers. The estimates for southern Bantu-speakers (SE Bantu-speakers, Sotho and Zulu) are larger than for the Zambian populations, except the BaTwa from Kafue (who have the largest estimate of all populations). The eastern Bantu-speakers, Luhya and Ba.Kiga, have lower (negative) estimates. As mentioned lines 280-285, we think that the different tests detect the same signal of non-west-African-like ancestry, and the high estimates for southern Bantu-speakers is likely due to Khoe-San ancestry in these populations.

- L. 378-383: I suggest that the authors report the male and female contribution from West Africans in Zambians, instead of that of KS HGs. This would avoid negative values; male contributions slightly above 1 may be rounded to 1.

Authors: We modified Tables S11 and S12 to report results from both the western-African and hunter-gather sources. We also modified the corresponding results section (lines 362-377) and discussion (rows 483-486) to focus more on the western-African ancestry, and suggested explanations for the negative values for the hunter-gatherer-like source.

- L. 775: I suggest the authors test 4 admixture events (in case RHG > Zambian BaTwa is supported).

Authors: Thank you for the suggestion. We ran `find_graphs` with four admixture edges for the dataset including the BaTwa from Kafue (following the same methodology as described previously). The fit with four admixture events is not significantly better than the fit with three admixture events ($p=0.87$). The graph with four admixture events does not include the RHGs.

- L. 125: Replace "resembles" by "resemble"

- L. 158: Remove the first "more"?

- L. 197: Replace "have" by "has"

- L. 312: Replace "that" by "who"

- L. 556: Replace "African" by "Africa"

- L.761: Replace "admixture" by "admixture"

- L. 770: Replace "follow" by "follows"

- L. 778-780: Replace "was" by "were" or add "model"

Authors: Thank you for noticing these typos. The text was modified accordingly.

- L. 891: Data and code are not made available yet.

Authors: The GitHub repository is now public and accessible at:

https://github.com/Gwennid/zambia_batwa/

The data is being checked by EGA for conformity purposes. The EGA accession number will be provided in the final version.

Reviewer #3 (Remarks to the Author):

In this revised manuscript, Breton et al. combined data genetic data from five Zambian populations (including Bangweulu and Kafue "BaTwa" populations) with data from 39 other African populations to infer demographic history and historical patterns of admixture. Note that I was not one of the original reviewers, and in reviewing this paper I have opted to avoid groupthink and take a fresh perspective (i.e., I did not focus on the response to reviewer comments). Overall, this is a solid paper that is well-suited for a journal like Nature Communications. The authors have done a thorough job in their population genetics analyses, and they have employed the appropriate computational approaches. The writing is easy to follow and most figures and tables are easy to interpret (see below for minor exceptions). On the whole, I liked this paper and feel that it makes a valuable contribution to the anthropological genetics literature. However, minor revisions are still required for it to be suitable for publication. These changes involve some changes in wording, double-checking the results shown in Figure 3, briefly addressing how SNP ascertainment may bias their results, and ensuring that data and code availability standards are upheld.

Additional comments and suggestions are listed below.

Abstract (page 3, lines 40-41): The authors should be careful about using phrase "does not match any living group" when referring to the hunter-gatherer ancestry component found in BaTwa genomes, especially in light of the Taíno genomics controversy that occurred over a decade ago. In addition, it would be good to include details about sample sizes in the abstract, particularly the number of novel samples that have been genotyped as part of this study.

Authors: Thank you for pointing this out. We removed the phrase "does not match any living group" as we already commented on the uniqueness of the genetic component (line 41). We added the sample sizes (lines 32-33).

Introduction: Multiple papers in the past three years have referred to the Sabue people of Ethiopia as the Chabu, and it might be good to mention this alternative label in this manuscript.

Authors: We added the alternative label at the first mention of the Sabue, line 205.

Figure 3 and lines 281-285: It is quite surprising that the patterns are so similar in panels A and B in this figure. Similarly, the values of Spearman's rho that are mentioned in lines 281-285 are striking enough to provoke a little skepticism (1.0 and -1.0). Why are these trends so similar for each of the populations described in Figures 3 and S6? The authors are encouraged to add a couple more sentences to this manuscript to help explain these striking patterns. Also, although alpha statistics from f4 ratio tests are appropriate, readers may be confused by negative values of "estimated fractions of Jul'hoansi/Baka/Hadza ancestry" in Figures 3 and S6.

Authors: Since it is a rank-correlation, it might not be that surprising and confirm the results (see the “logfile” files for the different tested populations at https://github.com/Gwennid/zambia_batwa/tree/main/results/revisions_NatureCommunication/f4ratio). We added a couple of sentences to explain the patterns (lines 280-285). We added a comment about the negative statistics in the legends of figure 3. We note that estimating the exact values of admixture fractions with f4 is reliable only if you have the correct source populations. Otherwise, it gives a relative picture, and that is what we see here. We added a comment in the figure text of Figures 3, S6 and S18.

Figure S6: The Hadza are from Tanzania, not Ethiopia, yes?

Authors: Thank you for noticing this mistake. We corrected the figure text (page 72).

Figure 4: What was the minimum RoH length used to generate this figure? 1Mb? In any case, these detail should be included in the figure legend.

Authors: Yes, the minimum length is 1 Mb. This is now included in the figure legend (page 57).

The supplemental figures that depict co-ancestry curves (Figures S8, S9, S10, S11, S12, and S13) are missing axis labels. It also might be good to expand the figure legends of these supplemental figures to assist the reader. As presently shown, it is not immediately clear how one should interpret these curves.

Authors: We expanded the legends of figures S8-S13 to define the axes and the different lines depicted. We also highlighted panels of interest.

Page 12, line 244: How much affinity? It would be good to quantify this.

Authors: We added the value of the f3 statistics and the Z-score for the two significant tests (lines 244-245).

Page 13, line 278: Should this sentence refer to admixture BETWEEN or INTO populations?

Authors: This is correct, the admixture could be between the eastern African and the Zambian populations, or from the eastern African into the Zambian populations. It is not possible to tell the direction based solely on f3 and f4 statistics. Since we are focusing on the history of Zambian populations, and to keep the text more readable, we chose to talk about the admixture “into” the Zambians.

Page 15, lines 307-315: Given that this manuscript uses data from the Illumina H3Africa array, Fst statistics are likely to be affected by ascertainment bias (particularly for populations like the Bangweulu and Kafue). The authors are encouraged to mention this point and include any relevant citations.

Authors: We added a comment about ascertainment bias (lines 586-589). While we used the H3Africa array for the Zambian populations, which should mitigate ascertainment bias, most of our analyses are restricted to variants also present on the Illumina Omni1 or Omni 2 arrays (due to merging with other populations). For these arrays, it is unlikely that the bias will be larger for the BaTwa than for other African populations (for example Khoe-San or rainforest hunter-gatherers).

Page 17, lines 378-380: It would be good to either report the actual amounts of X chromosome and autosome ancestry or include confidence intervals for female contributions (especially in light of the 2023 AJHG paper by Pfennig et al. about sex-biased admixture).

Authors: Thank you for this suggestion. We completed Table S12 with the average autosomal and X chromosomal ancestry proportions. We also added the ratios for the western-African-like source and modified the results (rows 358-373) and discussion (rows 478-482).

Please ensure that the EGA accession numbers are included in the final/next version of this manuscript. Also, the GitHub repository link listed on page 39 did not work.

Authors: the GitHub repository is now public and the link should work. The EGA accession numbers will be added to the final manuscript.

Reviewers' Comments:

Reviewer #3:

Remarks to the Author:

I am satisfied with both the response to reviewer comments and the edits that have been made to NCOMMS-23-11554B. In my opinion this manuscript is acceptable for publication and it meets the standards of Nature Communications. It is a solid anthropological genetics paper, and it will be good to see this comprehensive analysis of BaTwa genetics in the literature.